# Rethinking Multimodal Learning from the Perspective of Mitigating Classification Ability Disproportion

**Qing-Yuan Jiang**[†‡]**, Longfei Huang**[†]**, Yang Yang**[†*]

[†]Nanjing University of Science and Technology
[‡]State Key Lab. for Novel Software Technology, Nanjing University, P.R. China
`{jiangqy, hlf, yyang}@njust.edu.cn`

## Abstract

Multimodal learning (MML) is significantly constrained by modality imbalance, leading to suboptimal performance in practice. While existing approaches primarily focus on balancing the learning of different modalities to address this issue, they fundamentally overlook the inherent disproportion in model classification ability, which serves as the primary cause of this phenomenon. In this paper, we propose a novel multimodal learning approach to dynamically balance the classification ability of weak and strong modalities by incorporating the principle of boosting. Concretely, we first propose a sustained boosting algorithm in multimodal learning by simultaneously optimizing the classification and residual errors. Subsequently, we introduce an adaptive classifier assignment strategy to dynamically facilitate the classification performance of the weak modality. Furthermore, we theoretically analyze the convergence property of the cross-modal gap function, ensuring the effectiveness of the proposed boosting scheme. To this end, the classification ability of strong and weak modalities is expected to be balanced, thereby mitigating the imbalance issue. Empirical experiments on widely used datasets reveal the superiority of our method through comparison with various state-of-the-art (SOTA) multimodal learning baselines. The source code is available at `https://github.com/njustkmg/NeurIPS25-AUG`.

## 1 Introduction

In recent years, multimodal learning [33, 48, 43, 47, 21] has received growing attention for its ability to effectively integrate heterogeneous information. As extra information from multimodal data can be utilized, multimodal learning is expected to achieve better performance compared with unimodal approaches. However, contrary to expectations, multimodal learning has been surprisingly shown to underperform compared to unimodal ones in certain scenarios [42, 35, 46].

The root of this problem lies in the existence of the modality imbalance [42]. Concretely, different modalities in a joint-training paradigm typically converge at different speeds [35, 45]. The faster-converging modality, i.e., strong modality [49], tends to achieve higher performance, while the weak modality performs poorly. Subsequently, this disproportion in classification ability often leads to modality imbalance [42], ultimately resulting in lower performance.

Researchers have explored the modality imbalance issue from various perspectives in multimodal learning [42, 35, 52]. Given the inconsistent learning progress between strong and weak modalities, a natural idea [42, 35, 28, 50] is to manually intervene in their learning processes to achieve rebalancing. Another type of method is to bridge the information gap between modality training phases and enhance the interaction between different modalities during training. To be specific, impressive works [52, 11]

---

[*]Corresponding author.

39th Conference on Neural Information Processing Systems (NeurIPS 2025).

such as MLA [52], ReconBoost [21] and DI-MML [11] focus on bridging the learning gap of different modalities through injecting the optimization information between modalities.

Although the above methods can rebalance multimodal learning, they focus more on balancing the learning process while failing to enhance the classification ability explicitly. Compared to weaker modalities, stronger modalities typically yield more robust classifiers due to their more sufficient information [49]. Is there a way to directly improve the performance of weak classifiers to balance the classification performance between strong and weak modalities? A natural choice is boosting [13, 14], which utilizes the ensemble technique to enhance the ability of the weak classifier. We conduct a toy experiment to illustrate this idea on CREMAD dataset [4], where the classifier of weak modality is enhanced by the gradient boosting [14]. The results in Figure 1 present the comparison among naive MML, a model learning adjustment-based MML approach (G-Blend [42]), and gradient boosting-based MML (MML w/ GB). For MML w/ GB, we apply the gradient boosting algorithm to further improve the trained video model using naive MML, while keeping the audio model fixed. We can find that the classification gap between video and audio modalities of naive MML and G-Blend is relatively large. More importantly, for MML w/ GB, the accuracy of audio modality remains unchanged, but the accuracy of video is greatly improved, leading to the improvement of overall accuracy. This demonstrates the feasibility and effectiveness of using boosting to balance the classification ability of strong and weak modalities in mitigating modality imbalance.

According to the aforementioned observations, in this paper, we propose a novel multimodal learning approach by designing a sustained boosting algorithm to facilitate the classification ability of the weaker modality. Concretely, we first propose a sustained boosting algorithm by jointly optimizing the classification loss and residual error, aiming to enhance the classification performance of the weak modality. The algorithm takes features extracted by the encoder as input and provides predictions for the given data. Then, we employ the confident score [35] to monitor the learning status during joint training, and further propose an adaptive classifier assignment (ACA) strategy to adjust the classifier of weak modality. To this end, we can enhance the classification ability for the weak modality, thereby rebalancing the classification ability of strong and weak modalities. Meanwhile, we theoretically show that, under the boosting algo-

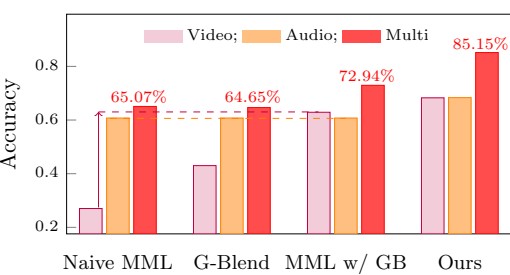

Figure 1: Comparison with naive MML, gradient boosting based MML (MML w/ GB), G-Blend [42], and Ours on CREMAD dataset. We find that enhancing the classification performance of the weak modality narrows the performance gap between the two modalities and improves overall performance.

rithm framework, the gap between different losses is provably convergent. In Figure 1, we present the classification enhancement results of our method (Ours). We can find that the performance of our method outperforms that of MML w/ GB thanks to the sustained boosting and adaptive classifier assignment strategy. Furthermore, it is worth mentioning that ReconBoost [21] also employs the gradient boosting algorithm for multimodal learning. However, unlike our approach, ReconBoost uses gradient boosting to iteratively learn complementary information across modalities. Our main contributions are outlined as follows:

- A novel sustained boosting algorithm in MML is proposed. This algorithm aims to simultaneously minimize the classification and residual errors to facilitate the classification ability of the weak modality.

- A novel adaptive classifier assignment strategy is proposed to dynamically enhance the classification ability of weak modality based on the learning status, thus rebalancing the classification ability of all modalities.

- We theoretically analyze the impact of the boosting algorithm on the loss gap between different modalities and prove its convergence.

- Experiments reveal that our approach can outperform SOTA baselines to achieve the best performance by a large margin on widely used datasets.

## 2 Related Work

### 2.1 Rebalanced Multimodal Learning

The goal of multimodal learning [48, 24, 33, 39, 22] is to fuse the multimodal information from diverse sensors. Compared to unimodal methods, multimodal learning can mine data information from different perspectives, thus the performance of multimodal learning should be better [30, 38, 20, 16]. However, due to heterogeneity of multimodal data, multimodal learning often encounters imbalance problems [42, 23] in practice, leading to performance degeneration of multimodal learning.

Early pioneering works [42, 35, 12, 17] focus more on adaptively adjusting the learning procedure for different modalities. Representative approaches in this category employ different learning strategies, e.g., gradient modulation [35, 28] and learning rate adjustment [50], to rebalance the learning of weak and strong modalities. Other approaches including MLA [52], ReconBoost [21] and IGM [25] take a different path, focusing on enhancing the interaction between modalities to address the modality imbalance problem. For example, MLA [52] designs an alternating algorithm to train different modalities iteratively. During the training phase, the interaction is enhanced by transferring the learning information between different modalities. ReconBoost [21] balances modality learning by leveraging gradient boosting to capture information from other modalities during interactive learning. IGM [25] employs a flat gradient modification strategy to enhance the interactive multimodal learning.

The aforementioned methods focus on rebalancing the learning process for weak and strong modalities while failing to explicitly facilitate the classification ability of the weak modality. In this paper, we aim to address the modality imbalance issue from facilitating the classification ability of weak modality and rebalancing the classification ability of weak and strong modalities.

### 2.2 Boosting Method

Boosting algorithm [13, 14, 29, 8, 37] is one of the most important algorithms in ensemble learning. The core idea of boosting is to integrate multiple learners to create a strong learner. Adaboost [13], one of the earliest boosting algorithms, adjusts the weights of incorrectly classified data points, giving more attention to the harder-to-classify examples in each iteration. Gradient boosting [14], on the other hand, builds models in a stage-wise fashion, minimizing a loss function through gradient descent. It iteratively refines the overall model by fitting the negative gradient of the loss function [32] with respect to the model's predictions.

The key advantage of boosting lies in its ability to improve model accuracy without requiring complex individual models. Therefore, boosting becomes the natural choice for improving the performance of weak classifiers.

## 3 Methodology

### 3.1 Multimodal Learning

For simplicity, we use two modalities, i.e., audio and video, for illustration. It is worth mentioning that our method can be easily adapted to cases with more than two modalities.

Assume that we have $N$ data points, each of which has audio and video modalities. Without loss of generality, we use $\boldsymbol{X} = \{(\boldsymbol{x}_i^a, \boldsymbol{x}_i^v)\}_{i=1}^N$ to denote the multimodal data, where $\boldsymbol{x}_i^a$ and $\boldsymbol{x}_i^v$ denote the $i$-th data point of audio and video, respectively. In addition, we are also given a category labels set $\boldsymbol{Y} = \{\ \boldsymbol{y}_i \mid \boldsymbol{y}_i \in \{0,1\}^K\}_{i=1}^N$, where $K$ denotes the number of category labels. Given the above training information $\boldsymbol{X}$ and $\boldsymbol{Y}$, the goal of multimodal learning is to train a model to fuse the multimodal information and predict its category label as accurately as possible.

For the sake of simplicity, we use superscript $o$ to indicate the module corresponding to a specific modality in this section, where $o \in \{a, v\}$. With the rapid growth of deep learning, representative MML approaches [33, 42, 12, 28] have adopted deep neural network (DNN) for multimodal learning. Following these methods, we also utilize DNN to construct our models. Specifically, we use $\phi^o(\cdot)$ to denote encoders. Then the features can be calculated by $\boldsymbol{u}^o = \phi^o(\boldsymbol{x}^o; \theta^o)$, where $\theta^o$ denotes the encoder parameters. Then, the prediction of given data can be calculated by a classifier $\psi^o(\cdot)$: $\boldsymbol{p}^o = \psi^o(\boldsymbol{u}^o; \Theta^o)$, where $\Theta^o$ denotes the parameters of the classifier. Based on $\boldsymbol{p}^o$ and its ground-

truth, the objective function can be written as:

$$\mathcal{L}_{\text{CE}}(\boldsymbol{X}^o, \boldsymbol{Y}) = \frac{1}{N} \sum_{i=1}^{N} \ell(\boldsymbol{p}_i^o, \boldsymbol{y}_i) = -\frac{1}{N} \sum_{i=1}^{N} \boldsymbol{y}_i^\top \log(\boldsymbol{p}_i^o), \tag{1}$$

where $\Phi^o \triangleq \{\theta^o, \Theta^o\}$ denotes the parameters to be learned, $\boldsymbol{X}^o \triangleq \{\boldsymbol{x}_i^o\}_{i=1}^N$ and $\ell(\cdot)$ denotes the cross entropy loss.

## 3.2 Sustained Boosting

By training the model for each modality based on objective function (1), we can obtain multiple individual classifiers. Due to the existence of strong and weak modalities [49], these classifiers exhibit different classification abilities. Hence, we can employ boosting technique [14] to improve the classification ability of weak modality.

Concretely, assuming the classification performance of the $o$-modality requires improvement, we first apply the gradient boosting algorithm to train $n$ classifiers for the $o$-modality. Since feature extraction focuses on common patterns, we set the encoders of all classifiers to be shared. Then the $j$-th classifier can be defined as: $\Phi_t^o \triangleq \{\theta^o, \Theta_t^o\}, t \in \{1, \cdots, n\}$. In practice, we adopt multiple fully-connected layers and nonlinear activation rectified linear unit (ReLU) [1] to construct our classification module. This module called the configurable classifier, is relatively independent and can be adjusted based on the classification ability. Furthermore, we adopt the shared head structure commonly used in MML [52, 7, 25] to strengthen the interaction between weak and strong modalities during training.

Inspired by gradient boosting [14], the classification ability can be facilitated through minimizing the residual error introduced by previous classifiers. Concretely, when we learn $t$-th classifier, the residual labels are defined as:

$$\hat{\boldsymbol{y}}_{it}^o = \boldsymbol{y}_i - \lambda \sum_{j=1}^{t-1} \boldsymbol{y}_i \odot \boldsymbol{p}_{ij}^o,$$

where $\lambda \in [0, 1]$ is used to soften hard labels [40], $\odot$ denotes the element-wise product, and we utilize $\boldsymbol{y}_i$ to mask non ground-truth labels to ensure the non-negativity of residual labels. Then the objective function can be defined as follows:

$$\epsilon(\boldsymbol{x}_i^o, \boldsymbol{y}_i, t) = \ell(\boldsymbol{p}_{it}^o, \hat{\boldsymbol{y}}_{it}^o), \tag{2}$$

where $\boldsymbol{p}_{it}^o$ denotes the prediction obtained by $t$-th classifier for $i$-th data point. Since we utilize a shared encoder, the encoder will be updated when training the $t$-th classifier. Therefore, other classifiers must be updated simultaneously to prevent performance degradation. The corresponding objective can be formed as:

$$\epsilon_{\text{all}}(\boldsymbol{x}_i^o, \boldsymbol{y}_i, t) = \ell\left(\boldsymbol{p}_{it}^o + \sum_{j=1}^{t-1} \boldsymbol{p}_{ij}^o, \boldsymbol{y}_i\right) = \ell\left(\sum_{j=1}^{t} \boldsymbol{p}_{ij}^o, \boldsymbol{y}_i\right). \tag{3}$$

Meanwhile, we have to ensure the first $t-1$ classifiers are well-trained. Hence, we define the following objective for $t-1$ classifiers:

$$\epsilon_{\text{pre}}(\boldsymbol{x}_i^o, \boldsymbol{y}_i, t) = \ell\left(\sum_{j=1}^{t-1} \boldsymbol{p}_{ij}^o, \boldsymbol{y}_i\right). \tag{4}$$

By combining (2), (3), and (4), the objective can be defined as:

$$L(\boldsymbol{x}_i^o, \boldsymbol{y}_i, t) = \epsilon(\boldsymbol{x}_i^o, \boldsymbol{y}_i, t) + \epsilon_{\text{all}}(\boldsymbol{x}_i^o, \boldsymbol{y}_i, t) + \epsilon_{\text{pre}}(\boldsymbol{x}_i^o, \boldsymbol{y}_i, t). \tag{5}$$

Unlike traditional gradient boosting [14], our method sustainedly minimizes classification and residual errors by optimizing (5). The loss function of sustained boosting can be formed as:

$$\mathcal{L}_{\text{SUB}}(\boldsymbol{X}^o, \boldsymbol{Y}, n^o; \Phi^o) = \frac{1}{N} \sum_{i=1}^{N} L(\boldsymbol{x}_i^o, \boldsymbol{y}_i, n^o), \tag{6}$$

where $n^o$ denotes the number of classifier for $o$-modality.

---

**Algorithm 1** Learning algorithm of our proposed method.

---

**Require:** Training data $\boldsymbol{X}$, category labels $\boldsymbol{Y}$.
**Ensure:** The learned DNN models for all modalities.
  1: **INIT** Initialize the number of classifier $n^a = 1$, $n^v = 1$. Initialize iteration $t = 1$. Initialize
      DNN parameters $\Phi_t^a$ and $\Phi_t^v$.
  2: **for** $t = 1 \mapsto \#iterations$ **do**
  3:     Sample a mini-batch $\boldsymbol{X}_t = \{(\boldsymbol{x}_i^a, \boldsymbol{x}_i^v)\}_{i=1}^{n_b}$;                         ▷ Learn MML models.
  4:     $\forall \boldsymbol{x}_i^a, \boldsymbol{x}_i^v \in \boldsymbol{X}_t$, calculate features $\boldsymbol{u}_i^a$ and $\boldsymbol{u}_i^v$;
  5:     Calculate predictions $\{\boldsymbol{p}_{ij}^a\}_{j=1}^{n^a}$ and $\{\boldsymbol{p}_{ij}^v\}_{j=1}^{n^v}$.
  6:     Calculate loss in (6) based on predictions;
  7:     Update DNN parameters $\Phi_t^a$ and $\Phi_t^v$ based on SGD;
  8:     **if** $\mathrm{mod}(t, t_N) = 0$ **then**               ▷ Adaptive Classification assignment strategy.
  9:         Calculate confident score $\{s_t^a, s_t^v\}$ based on predictions;
10:         **if** $s_t^a - \sigma s_t^v > \tau$ **then**
11:             Add a classifier for audio modality;
12:             $n^a = n^a + 1$;
13:         **else if** $s_t^a - \sigma s_t^v < \tau$ **then**
14:             Add a classifier for video modality;
15:             $n^v = n^v + 1$;
16:         **end if**
17:     **end if**
18: **end for**

---

## 3.3   Adaptive Classifier Assignment

Thus far, we have defined a configurable classifier module and designed a sustained boosting in multimodal learning to enhance the classification performance of weak modality. However, recent studies [35] have shown that differences between modalities evolve dynamically due to imbalance issues in multimodal learning. This implies the need to design a strategy for enhancing classification ability that adapts to dynamic changes. Hence, we propose an adaptive classifier assignment strategy to adjust the number of the weak classifier. For simplicity, we redefine the modality classifiers as: $\Phi_t^o \triangleq \{\theta^o, \Theta_t^o\}, t \in \{1, \cdots, n^o\}$, where $n^o$ is the parameter to be updated.

Then, we utilize confident score to monitor the learning status. At $t$-th iteration, confident score can be calculated by:

$$\forall o \in \{a, v\}, s_t^o = \frac{1}{N} \sum\nolimits_{i=1}^N \boldsymbol{y}_i^\top \left[ \sum\nolimits_{j=1}^{n^o} \boldsymbol{p}_{ij}^o \right].$$

The confident score reflects the classification ability of the models. Hence, if $s_t^a - \sigma s_t^v > \tau$, we assign a new configurable classifier for video modality at this iteration, where $\sigma \geq 1$ is the coefficient. $\tau$ is the dead zone for fault tolerance. On the contrary, we also assign a new configurable classifier for audio modality if $s_t^a - \sigma s_t^v < \tau$.

Our algorithm is summarized in Algorithm (1). In practice, we perform adaptive classification assignment strategy to determine if we need to adjust the classification ability every $t_N$ iterations.

## 3.4   Theoretical Analysis

The sustained boosting algorithm is introduced to reduce the loss gap between different modalities. In this section, we theoretically analyze its effect on minimizing this gap. We first define the gap function as follows:

$$\mathcal{G}(\Phi) = \mathcal{L}^a(\Phi^a) - \mathcal{L}^v(\Phi^v), \tag{7}$$

where $\Phi^a$ and $\Phi^v$ respectively denote the parameters of audio and video modality, $\mathcal{L}^a$ and $\mathcal{L}^v$ denote the overall loss function $\mathcal{L}_{\mathrm{SUB}}$ for audio and video, respectively. Without loss of generality, we assume that $\mathcal{L}^a > \mathcal{L}^v$; the case where $\mathcal{L}^a < \mathcal{L}^v$ can be analyzed analogously.

We derive the following conclusions:

**Theorem 1 (Convergence of Gap Loss, Informal)** *Under some assumptions for the loss function and the effectiveness of sustained boosting algorithm, we have:*

$$\mathcal{G}(\Phi(T)) \leq \frac{1}{1 + \frac{\nu^2 \kappa^2}{2 L_a \beta^2} T \mathcal{G}(\Phi(0))} \mathcal{G}(\Phi(0)) \tag{8}$$

*where $\nu$, $\kappa$, $L_a$ and $\beta$ are constant.*

The results in Theorem 1 indicate that, by employing the gradient boosting algorithm, the loss gap between modalities converges at a rate of $\mathcal{O}(1/T)$. The proof can be found in the appendix.

## 4 Experiments

### 4.1 Experimental Setup

**Dataset:** We carry out the experiments on six extensive multimodal datasets, i.e., CREMAD [4], KSounds [2], NVGesture [31], VGGSound [6], Twitter [51], and Sarcasm [3] datasets. The CREMAD, KSounds, and VGGSound datasets consist of audio and video modalities. NVGesture dataset contains three modalities, i.e., RGB, optical flow (OF), and Depth. Twitter and Sarcasm datasets consist of image and text modalities.

To ensure a fair comparison, we strictly follow the data partitioning strategy adopted by representative methods [42, 35, 52, 21]. Specifically, the CREMAD dataset contains 7,442 clips, which are divided into training set with 6,698 samples and testing set with 744 samples. For KSounds dataset, which contains 19,000 video clips, is divided into training set with 15,000 clips, validation set with 1,900 clips, and testing set with 1,900 clips. VGGSound dataset includes 168,618 videos for training and validation, and 13,954 videos for testing. The NVGesture dataset is divided into 1,050 samples for training and 482 samples for testing. Twitter dataset is divided into training set with 3,197 pairs, validation set with 1,122 pairs and testing set with 1,037 pairs. Sarcasm dataset includes 19,816 pairs for the training set, 2,410 pairs for the validation set, and 2,409 pairs for the testing set. More details are provided in the appendix.

**Baselines:** We select two categories of methods for comparison, i.e., traditional multimodal fusion methods and rebalanced multimodal learning methods. In detail, traditional multimodal fusion methods include concatenation fusion (Concat), affine transformation fusion (Affine) [36], multi-layers lstm fusion (ML-LSTM) [34], prediction summation fusion (Sum) [48], and prediction weighting fusion (Weight) [48]. Rebalanced multimodal learning methods include MSES [15], G-blend [42], MSLR [50], OGM [35], PMR [12], AGM [28], MMParato [44], SMV [43], MLA [52], DI-MML [11], LFM [47], ReconBoost [21].

**Evaluation Protocols:** Following the setting of MLA [52] and ReconBoost [21], we adopt accuracy, mean average precision (MAP) and MacroF1 as evaluation metrics. The accuracy measures the proportion of correct predictions of total predictions. MAP returns the average precision of all samples. And MacroF1 calculates the average F1 across all categories.

**Implementation Details:** Following OGM [35], we employ ResNet18 [19] as the backbone to encode audio and video for CREMAD, KSounds and VGGSound datasets. All the parameters of the backbone are randomly initialized. For NVGesture dataset, we employ the I3D [5] as unimodal branch following the setting of [45]. We initialize the encoder with the pre-trained model trained on ImageNet. For the architecture of the configurable classifier, we explore a two-layer network, which can be denoted as "Layer1$(D \times 256) \mapsto$ ReLU $\mapsto$ Layer2 $(256 \times K)$". Here, $D$ denotes the output dimensions of encoders, "Layer1"/"Layer2" are fully connected layer, and "ReLU" denotes the ReLU [1] activation layer. Furthermore, the Layer2 is utilized as shared head for all modalities as described in Section 3. Both Layer1 and Layer2 are randomly initialized. In addition, all hyper-parameters are selected by using the cross-validation strategy. Specifically, we use stochastic gradient descent (SGD) as the optimizer with a momentum of 0.9 and weight decay of $1 \times 10^{-4}$. The initial learning rate is set to be $1 \times 10^{-2}$ for CREMAD, KSounds, VGGSound , and NVGesture datasets. During training, the learning rate is progressively reduced by a factor of ten upon observing loss saturates. The batch size is set to be 64 for CREMAD and KSounds datasets, 16 for VGGSound dataset, and 2 for NVGesture dataset. We set the iteration $t_N$ for checking whether to assign the classifier to 20 epochs for CREMAD, 5 for Twitter, 1 for Sarcasm, and 10 for VGGSound, KSounds, NVGesture datasets.

Table 1: Classification performance comparison with SOTA baselines. The best and second-best results are highlighted in **bold** and underline, respectively. The results with gray background denote the performance based on multimodal learning is inferior to that of the best unimodal approach.

| Method | CREMAD | | KSounds | | VGGSound | | Twitter | | Sarcasm | | NVGesture | |
|---|---|---|---|---|---|---|---|---|---|---|---|---|
| | Acc. | MAP | Acc. | MAP | Acc. | MAP | Acc. | F1 | Acc. | F1 | Acc. | F1 |
| Unimodal-1 | .4583 | .5879 | .5412 | .5669 | .4655 | .4701 | .5863 | .4333 | .7181 | .7073 | .7822 | .7833 |
| Unimodal-2 | .6317 | .6861 | .5562 | .5837 | .3494 | .3478 | .7367 | .6849 | .8136 | .8056 | .7863 | .7865 |
| Unimodal-3 | N/A | N/A | N/A | N/A | N/A | N/A | N/A | N/A | N/A | N/A | .8154 | .8183 |
| Concat | .6361 | .6841 | .6455 | .7130 | .5116 | .5352 | .7011 | .6386 | .8286 | .8240 | .8237 | .8270 |
| Affine | .6626 | .7193 | .6424 | .6931 | .5001 | .5155 | .7203 | .5992 | .8240 | .8188 | .8278 | .8281 |
| ML-LSTM | .6290 | .6473 | .6394 | .6902 | .4966 | .5139 | .7068 | .6564 | .8277 | .8205 | .8320 | .8330 |
| Sum | .6344 | .6908 | .6490 | .7103 | .5136 | .5338 | .7300 | .6661 | .8294 | .8247 | .8050 | .8067 |
| Weight | .6653 | .7134 | .6533 | .7110 | .5144 | .5300 | .7242 | .6516 | .8265 | .8219 | .7842 | .7939 |
| MSES | .6546 | .7138 | .6591 | .7196 | .4891 | .5429 | .7252 | .6439 | .8423 | .8369 | .8112 | .8147 |
| G-blend | .6465 | .7392 | .6722 | .7274 | .5086 | .5555 | .7309 | .6799 | .8286 | .8215 | .8299 | .8305 |
| MSLR | .6868 | .7412 | .6756 | .7282 | .4987 | .5415 | .7232 | .6382 | .8439 | .8378 | .8237 | .8284 |
| OGM | .6612 | .7372 | .6582 | .7159 | .4829 | .4978 | .7058 | .6435 | .8360 | .8293 | – | – |
| PMR | .6659 | .7058 | .6675 | .7274 | .4647 | .4866 | .7357 | .6636 | .8310 | .8256 | – | – |
| AGM | .6733 | .7807 | .6791 | .7388 | .4711 | .5198 | .7261 | .6502 | .8306 | .8293 | .8279 | .8284 |
| MMParato | .7487 | .8535 | .7000 | .7850 | .5125 | .5473 | .7358 | .6729 | .8348 | .8284 | .8382 | .8424 |
| SMV | .7872 | .8417 | .6900 | .7426 | .5031 | .5362 | .7428 | .6817 | .8418 | .8368 | .8352 | .8341 |
| MLA | .7943 | .8572 | .7004 | .7945 | .5165 | .5473 | .7352 | .6713 | .8426 | .8348 | .8340 | .8372 |
| DI-MML | .8158 | .8592 | .7203 | .7426 | .5173 | .5479 | .7248 | .6686 | .8411 | .8315 | – | – |
| LFM | .8362 | .9006 | .7253 | .7897 | .5274 | .5694 | .7501 | **.7057** | .8497 | .8457 | .8436 | .8468 |
| ReconBoost | .7557 | .8140 | .6855 | .7662 | .5097 | .5387 | .7442 | .6832 | .8437 | .8317 | .8386 | .8434 |
| Ours | **.8515** | **.9103** | **.7263** | **.7901** | **.5301** | **.5826** | **.7512** | .6962 | **.8510** | **.8458** | **.8501** | **.8533** |
| | ±0.0027 | ±0.0014 | ±0.0031 | ±0.0063 | ±0.0006 | ±0.0017 | ±0.0068 | ±0.0016 | ±0.0054 | ±0.0047 | ±0.0025 | ±0.0031 |

For all datasets, we search $\lambda$ in $\{0.1, 0.2, 0.33, 0.5, 1.0\}$. For all datasets, $\sigma$ and $\tau$ are set to be $1.0$ and $0.01$, respectively. For Twitter and Sarcasm dataset, following [51, 3], we adopt BERT [9] as the text encoder and ResNet50 [19] as the image encoder. We use Adam [26] as the optimizer, with an initial learning rate of $2 \times 10^{-5}$. The batch size is set to 32 for Twitter and Sarcasm datasets. The other parameter settings are the same as audio-video datasets. For comparison methods, the source codes of all baselines are kindly provided by their authors. For fair comparison, all baselines also adopt the same backbone and initialization strategy for the experiment. All experiments are conducted on an NVIDIA GeForce RTX 4090 and all models are implemented with pytorch.

## 4.2 Main Results

**Classification Performance Comparison:** The classification results on all datasets are reported in Table 1, where "$-$" denotes that corresponding methods cannot applied to the dataset with more than two modalities. And Unimodal-1/2/3 is used to denote the results based on unimodal. Unimodal-1/2 respectively denote the video/audio for CREMAD and KSounds, and text/image for Twitter and Sarcasm. Unimodal-1/2/3 denotes the RGB/OF/Depth modality for NVGesture dataset, respectively. Furthermore, the results with a gray background indicate that the performance based on multimodal learning is inferior to that of the best unimodal approach. We can draw the following observations: (1). Compared with unimodal baselines, traditional multimodal fusion methods and rebalanced multimodal learning methods can achieve better performance in almost all cases; (2). Our method can outperform existing SOTA baselines to achieve the best accuracy in all cases for multimodal situations; (2). The accuracy on NVGesture dataset demonstrates that our method can extend to the case with more than two modalities and achieve the best performance.

## 4.3 Sensitivity to Hyperparameter

We explore the influence of $\sigma$ and $\lambda$ on CREMAD and KSounds datasets in this section. More results can be found in appendix.

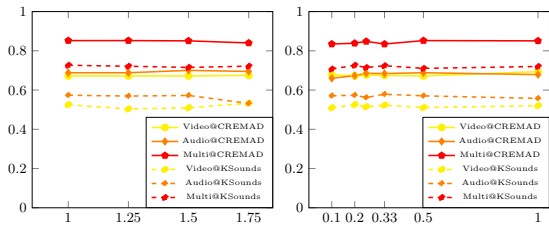

Figure 2: Sensitivity to $\sigma$ (left) and $\lambda$ (right).

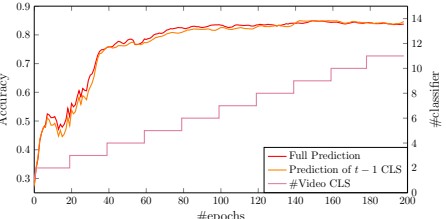

Figure 3: Performance comparison.

Table 2: The accuracy (Multi/Audio/Video) for ablation study on CREMAD and KSounds datasets.

| $\epsilon$ | $\epsilon_o$ | $\epsilon_p$ | CREMAD | KSounds |
|---|---|---|---|---|
| ✓ | ✗ | ✓ | 0.8333/0.6465/0.6734 | 0.6942/**0.5303**/0.5044 |
| ✗ | ✓ | ✗ | 0.8320/0.6573/0.6707 | 0.7190/**0.5303**/0.5145 |
| ✗ | ✓ | ✓ | 0.8360/**0.6841**/0.6371 | 0.7198/0.5261/0.5238 |
| ✓ | ✓ | ✓ | **0.8515**/0.6835/**0.6828** | **0.7263**/0.5257/**0.5736** |

Table 3: The impact of weak classifier selection strategy.

| Strategy | #Classifier | | Accuracy | | |
|---|---|---|---|---|---|
| | Audio | Video | Multi | Audio | Video |
| Fixed | 1 | 10 | 0.8091 | 0.6774 | 0.6156 |
| Fixed | 1 | 12 | 0.8118 | 0.6519 | 0.6277 |
| Adaptive | 1 | 10 | **0.8515** | **0.6835** | **0.6828** |

**Sensitivity to Threshold $\sigma$:** We study the influence of threshold $\sigma$ on CREMAD and KSounds datasets. The accuracy with different $\sigma \in [1, 1.75]$ is shown in Figure 2. We can find that our method is not sensitive to threshold $\sigma$ in a large range.

**Sensitivity to Smoothing Factor $\lambda$:** We explore the influence of smoothing factor $\lambda$ on CREMAD and KSounds datasets. The accuracy with different $\lambda \in [0.1, 1]$ is reported in Figure 2. We can find that our method is not sensitive to hyper-parameter smoothing factor $\lambda$ in a large range.

### 4.4 Ablation Study

We investigate the effectiveness of our method by analyzing the influence of the key components of our objectives in Equation (2), (3), and (4), respectively denoted as $\epsilon$, $\epsilon_o$, and $\epsilon_p$. The accuracy results on CREMAD and KSounds datasets are reported in Table 2. From Table 2, we can find that: (1). Both objectives in Equation (2), (3), and (4) can boost multimodal performance; (2). While the unimodal performance of the method using all objectives may not always reach the highest level, it achieves a more balanced classification performance across modalities. More results are reported in appendix.

We further investigate the impact of residual learning on classification performance by comparing the performance of all $t$ classifiers with that of the first $t - 1$ classifiers during the training. The results are presented in Figure 3, where the former accuracy is denoted as "Full Prediction" and the latter is denoted as "Prediction of $t - 1$ CLS". In Figure 3, we also present the number of the video classifier. We observe that the number of classifiers for the video modality has increased, and the performance of all $t$ classifiers is generally superior to that of the first $t - 1$ classifiers. This performance gain arises from our learning of the residual objective.

### 4.5 Further Analysis

**Impact of Weak Classifier Assignment Strategy:** We conduct an experiment to study the influence of adaptive classifier assignment strategy. Specifically, we design a fixed classifier assignment strategy for comparison. This approach allocates $n^{(\text{fix})}$ classifiers for weak modality during the init stage. And we no longer dynamically adjust the number of classifiers during training for weak modality.

The results on CREMAD dataset are reported in Table 3, where $n^{(\text{fix})}$ is set to be 10 and 12. The results in Table 3 demonstrate that our proposed adaptive classifier assignment strategy can boost performance compared with fixed classifier strategy. This is because our method dynamically adjusts modality classification performance in response to modality imbalance during training.

**Visualization Results:** We further study the property of embeddings through visualization. Specifically, we illustrate the t-SNE [41] results on CREMAD dataset for naive multimodal learning (naive

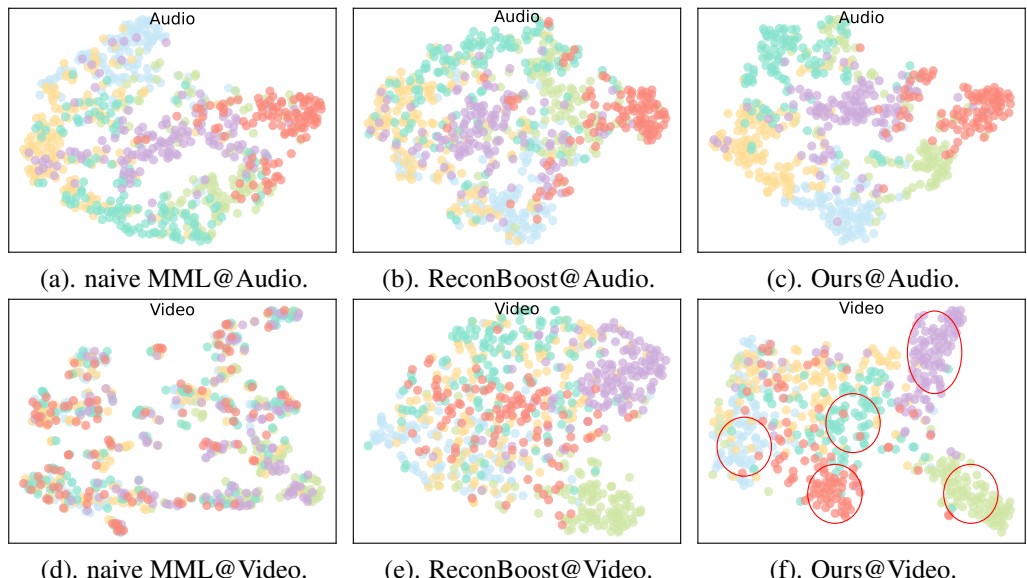

(a). naive MML@Audio.    (b). ReconBoost@Audio.    (c). Ours@Audio.

(d). naive MML@Video.    (e). ReconBoost@Video.    (f). Ours@Video.

Figure 4: Visualization on CREMAD dataset. The video visualization highlights the need to improve weak modality classification.

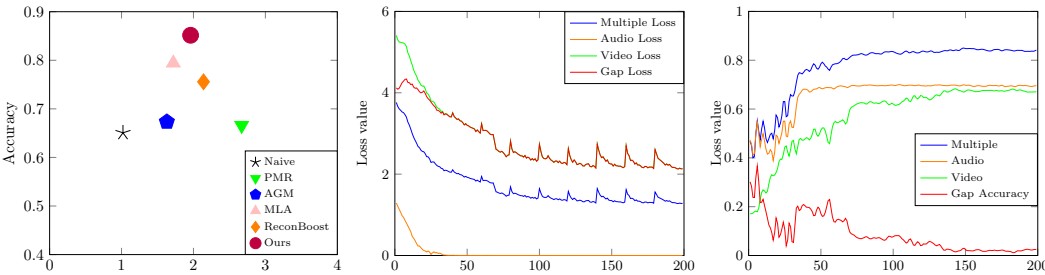

Figure 5: Training time (hrs).    Figure 6: Loss visualization.    Figure 7: Accuracy change.

MML), ReconBoost [21], and our method in Figure 4. From Figure 4, we can find that: (1). Compared to naive MML, our method and ReconBoost can learn more discriminative multimodal features, as both approaches enhance the weak modality using information from the strong modality; (2). Compared to ReconBoost, our method demonstrates significantly superior classification performance on the video modality, with several distinct categories highlighted by circle markers in Figure 4 (f). This improvement is primarily attributed to our explicit enhancement of the classification capabilities of the weaker modality.

**Training Overhead:** Considering that our method introduces additional computational overhead due to the use of the sustained boosting strategy, we compare its training cost with that of competitive SOTA baselines, including Naive, PMR, AGM, MLA, and ReconBoost, through empirical experiments under the same setting. The results are shown in Figure 5. It can be observed that our method achieves the best accuracy while maintaining competitive training time.

**Convergence of Gap Loss:** To further validate the convergence of the gap function, we conducted experiments on the CREMAD dataset. The change of gap function $\mathcal{G}(\Phi)$ during training process is reported in Figure 6, where we also report the unimodal loss and multimodal loss. It can be observed that as training progresses, the gap function gradually converges. Moreover, we report the accuracy changes during training in Figure 7. From Figure 7, it can be seen that the accuracy exhibits a similar convergence trend.

**Impact of the Model Capacity:** Since our method utilizes multiple designed classifiers for weak modality, it essentially utilizes more model parameters than baselines. Does our method benefit solely from having more model parameters? We conduct an experiment to verify this. Specifically, we

Table 4: The impact of model capacity.

| Method | Arch. | | #params. | | Acc. |
|---|---|---|---|---|---|
| | Audio | Video | Audio | Video | |
| Unimodal | - | R18 | - | 11.8M | 0.4718 |
| Unimodal | - | R34 | - | 23.3M | 0.4731 |
| Naive | R18 | R18 | 11.8M | 11.8M | 0.6507 |
| Naive | R18 | R34 | 11.8M | 23.3M | 0.6277 |
| Ours | R18 | R18 | 11.8M | 12.8M | **0.8515** |

Table 5: Performance comparison in scenario with modality missing.

| Method | $rate = 0\%$ | $rate = 20\%$ | $rate = 50\%$ |
|---|---|---|---|
| Naive MML | 0.6507 | 0.5849 | 0.5242 |
| ReconBoost | 0.7557 | 0.6321 | 0.5568 |
| MLA | 0.7943 | 0.6935 | 0.5753 |
| Ours | **0.8515** | **0.7540** | **0.6008** |

replace backbone ResNet18 (R18) of weak modality video as a larger backbone, i.e., ResNet34 (R34). Then we run the baselines and our method on CREMAD dataset. The results are shown in Table 4. From Table 4, we can find that our approach improves accuracy by nearly 20% with only an additional 1M parameters compared to naive MML with the same network architecture. Our method also outperforms the naive MML baseline with larger backbone. Furthermore, we observe an interesting and counterintuitive phenomenon. That is, the method with ResNet34 is worse than that with ResNet18. The reason behind this may be that the ResNet34-based method is more difficult to converge.

**Stability under Modality Missing:** Our method is adaptable to scenarios involving missing modalities. We comprehensively evaluate its performance under test-time missing modality conditions. Specifically, test-time missing refers to cases where the modalities are complete during training but missing during the testing phase. The experiments are conducted on CREMAD dataset with different missing rate [52]. Naive MML, ReconBoost [21], and MLA [52] are selected as baselines for comparison, where MLA introduces specifically designed algorithms to address the corresponding challenges in modality missing. We report the results with missing rate 20% and 50% in Table 5. We can find that as the modality missing rate increases, the performance of all methods declines. Nevertheless, our method consistently achieves the best performance under all missing rates, demonstrating the effectiveness of our method in scenario with missing modality.

## 5   Conclusion

To address the modality imbalance issue, we propose a novel multimodal learning approach by designing a sustained boosting algorithm to dynamically enhance the classification ability of weak modality. Concretely, we first propose a sustained boosting algorithm for multimodal learning by minimizing the classification and residual errors simultaneously. Then, we propose an adaptive classifier assignment strategy to dynamically facilitate the classification ability of weak modality. The effectiveness of the proposed boosting algorithm is theoretically guaranteed by analyzing the convergence properties of the cross-modal gap function. To this end, the classification ability can be rebalanced adaptively during the training procedure. Experiments on widely used datasets reveal that our proposed method can achieve state-of-the-art performance compared with various baselines by a large margin.

**Limitations:** Our proposed method mainly focuses on the classifier of each modality. For early fusion MML, our method can extend to balance the strong, weak, and fusion classification abilities. In addition, our theoretical analysis only examines the effect of the boosting algorithm on the convergence of the cross-modal gap function. In the full iterative framework, the overall convergence behavior and its influence on the model's learning capability warrant further investigation. We leave a more comprehensive investigation as future work.

## Acknowledgments and Disclosure of Funding

This work was supported in part by the National Key R&D Program of China (2022YFF0712100), in part by the NSFC (62276131, 62506168), in part by the Natural Science Foundation of Jiangsu Province of China under Grant (BK20240081, BG2024042, BK20251431), and in part by the Fundamental Research Funds for the Central Universities (No.30925010205).

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

# A   Proof

During training process, the gradient boosting strategy will be applied when the performance gap is larger than a threshold at each $c$ iterations. Now we analyze the convergence rate of $\mathcal{G}$ with gradient boosting strategy.

Furthermore, we make the following assumption:

**Assumption 1 (Strongly Convexity)**  *We assume that the function $\{\mathcal{L}^a(\cdot), \mathcal{L}^v(\cdot)\}$ is $\mu$-strongly convex. That is, for any $\Phi$ we have:*

$$\forall o \in \{a, v\}, \ \mathcal{L}^o(\Phi^o) \geq \mathcal{L}^o(\Phi^*) + \frac{\mu_o}{2}\|\Phi^o - \Phi^*\|^2 \tag{9}$$

*and*

$$\forall o \in \{a, v\}, \ \|\nabla\mathcal{L}^o(\Phi(t))\|^2 \geq \mu_o|\mathcal{L}^o(\Phi(t)) - \mathcal{L}^o(\Phi^*)| \tag{10}$$

**Assumption 2 (Smoothness)**  *We assume that the functions $\{\mathcal{G}(\cdot), \mathcal{L}_a(\cdot), \mathcal{L}_v(\cdot)\}$ are $L$-smooth. That is, for any $\Phi, \Phi'$, we have*

$$\forall o \in \{a, v\}, \ \mathcal{L}^o(\Phi) - \mathcal{L}^o(\Phi') \leq \langle\nabla\mathcal{L}^o(\Phi'), \Phi - \Phi'\rangle + \frac{L_o}{2}\|\Phi - \Phi'\|^2. \tag{11}$$

*and*

$$\mathcal{G}(\Phi) - \mathcal{G}(\Phi') \leq \langle\nabla\mathcal{G}(\Phi'), \Phi - \Phi'\rangle + \frac{L_g}{2}\|\Phi - \Phi'\|^2. \tag{12}$$

**Assumption 3 (Effectiveness of Weak Classifier)**  *We assume that the gradient boosting strategy for weak modality is effective. That is, there exists a $\nu \in (0, 1)$:*

$$\langle\nabla\mathcal{L}^a(\Phi(t)), h(\Phi(t))\rangle \leq -\nu\|\nabla\mathcal{L}^a(\Phi(t))\|^2. \tag{13}$$

*And there exists a constant $\beta > 0$:*

$$\|h(\Phi(t))\| \leq \beta\|\nabla\mathcal{L}^a(\Phi(t))\|, \tag{14}$$

*where $h(\cdot)$ denotes the function obtained by boosting algorithm. $h(\cdot)$ aims to fit the negative gradient of weak modality model.*

Based on the definition of $h(\cdot)$, the updating rule can be formed as:

$$\Phi(t+1) = \Phi(t) - \eta\nabla\mathcal{L}(\Phi(t)) \tag{15}$$
$$= \Phi(t) + \eta h(\Phi(t)) \tag{16}$$

where $\eta$ denotes the learning rate.

**Assumption 4 (Optimal of Multimodels)**  *We assume that there exists an optimal solution $\Phi^*$ so that we have: $\nabla\mathcal{L}^a(\Phi^*) = \nabla\mathcal{L}^v(\Phi^*) = 0$.*

**Assumption 5 (Bounded Initial Value of $\mathcal{G}(\Phi(t))$)**  *We assume that the initial value of $\mathcal{G}(\Phi(t))$ is bounded, i.e., $\mathcal{G}(\Phi(0)) \leq \infty$.*

Based on Assumption (1), (4), we have the following lemma:

**Lemma 1 (Gap Bound)**  *There exists a constant $\kappa$ such that the loss gap function $\mathcal{G}(\cdot)$ and the gradient norm of the strong modality satisfy the following relationship:*

$$\|\nabla\mathcal{L}^a(\Phi_t)\| \geq \kappa|\mathcal{G}(\Phi_t)|. \tag{17}$$

**Proof 1 (Proof of Lemma 1)**  *Since $\mathcal{L}^a(\cdot)$ and $\mathcal{L}^v(\cdot)$ are strongly convex, there exists an constant $\rho \in (0, 1)$ satisfying:*

$$\|\Phi(t) - \Phi^*\| \leq \rho^t\|\Phi(0) - \Phi^*\| \tag{18}$$

*Since the smoothness, we have:*

$$\forall o \in \{a, v\}, \mathcal{L}^o(\Phi(t)) - \mathcal{L}^o(\Phi^*) \leq \frac{L^o}{2}\|\Phi(t) - \Phi^*\|^2 \tag{19}$$

$$\leq \frac{L^o}{2}\rho^{2t}\|\Phi(0) - \Phi^*\|^2 \tag{20}$$

*Then we have:*

$$\mathcal{G}(\Phi(t)) = \mathcal{L}^a(\Phi(t)) - \mathcal{L}^v(\Phi(t)) \tag{21}$$

$$= \mathcal{L}^a(\Phi(t)) - \mathcal{L}^a(\Phi^*) - (\mathcal{L}^v(\Phi(t)) - \mathcal{L}^v(\Phi^*)) \tag{22}$$

$$\leq |\mathcal{L}^a(\Phi(t)) - \mathcal{L}^a(\Phi^*)| + |(\mathcal{L}^v(\Phi(t)) - \mathcal{L}^v(\Phi^*))| \tag{23}$$

$$\leq \left(\frac{L_a}{2} + \frac{L_v}{2}\right)\rho^{2t}\|\Phi(0) - \Phi^*\|^2 \tag{24}$$

$$\leq \left(\frac{L_a}{2} + \frac{L_v}{2}\right)\|\Phi(0) - \Phi^*\|^2 \tag{25}$$

*By setting $c = \left(\frac{L_a}{2} + \frac{L_v}{2}\right)\|\Phi(0) - \Phi^*\|^2$, we have:*

$$|\mathcal{G}(\Phi(t))| \leq c \tag{26}$$

*Then we have:*

$$|\mathcal{G}(\Phi(t))| \geq \frac{1}{c}|\mathcal{G}(\Phi(t))|^2 \tag{27}$$

*According to Assumption (1), we have:*

$$\|\nabla\mathcal{L}^a(\Phi(t))\|^2 \geq \mu_a|\mathcal{L}^a(\Phi(t)) - \mathcal{L}^a(\Phi^*)| \tag{28}$$

*Suppose that two modalities achieve the same optimal value, i.e., $\mathcal{L}^a(\Phi^*) = \mathcal{L}^v(\Phi^*)$. Then we have:*

$$\|\nabla\mathcal{L}^a(\Phi(t))\|^2 \geq 2\mu|\mathcal{L}^a(\Phi(t)) - \mathcal{L}^a(\Phi^*)| \tag{29}$$

$$= 2\mu|\mathcal{L}^a(\Phi(t)) - \mathcal{L}^v(\Phi^*)| \tag{30}$$

$$\geq 2\mu|\mathcal{L}^a(\Phi(t)) - \mathcal{L}^v(\Phi(t))| \tag{31}$$

$$= 2\mu|\mathcal{G}(\Phi(t))| \tag{32}$$

$$\geq \frac{2\mu}{c}|\mathcal{G}(\Phi(t))|^2 \tag{33}$$

*By setting $\kappa = \sqrt{\frac{2\mu}{c}}$, we have:*

$$\|\nabla\mathcal{L}^a(\Phi(t))\| \geq \kappa|\mathcal{G}(\Phi(t))| \tag{34}$$

We have the following theorem:

**Theorem 1 (Convergence of $\mathcal{G}$ with Gradient Boosting)** *Under assumption (2) and (3), if the learning rate is set as $\eta_t = \frac{\nu}{L_a\beta^2}$, we have:*

$$\mathcal{G}(\Phi(T)) \leq \frac{\mathcal{G}(\Phi(0))}{1 + dT\mathcal{G}(\Phi(0))} = \mathcal{O}(\frac{1}{T}), \tag{35}$$

*where $d = \frac{\nu^2\kappa^2}{2L_a\beta^2}$.*

**Proof 2 (Proof of Theorem 1)** *Based on smooth assumption, we have:*

$$\mathcal{L}^a(\Phi(t+1)) \leq \mathcal{L}^a(\Phi(t)) + \langle\nabla\mathcal{L}^a(\Phi(t)), \Phi(t+1) - \Phi(t)\rangle + \frac{L_a}{2}\|\Phi(t+1) - \Phi(t)\|^2, \tag{36}$$

$$= \mathcal{L}^a(\Phi(t)) + \eta_t\langle\nabla\mathcal{L}^a(\Phi(t)), h(\Phi(t))\rangle + \frac{L_a\eta_t^2}{2}\|h(\Phi(t))\|^2, \tag{37}$$

$$\overset{(13)}{\leq} \mathcal{L}^a(\Phi(t)) - \nu\eta_t\|\nabla\mathcal{L}^a(\Phi(t))\|^2 + \frac{L_a\eta_t^2}{2}\|h(\Phi(t))\|^2, \tag{38}$$

$$\overset{(14)}{\leq} \mathcal{L}^a(\Phi(t)) - \nu\eta_t\|\nabla\mathcal{L}^a(\Phi(t))\|^2 + \frac{L_a\eta_t^2\beta^2}{2}\|\nabla\mathcal{L}^a(\Phi(t))\|^2 \tag{39}$$

Considering that the parameters for each modality are independent, we have: $\mathcal{L}^v(\Phi(t+1)) = \mathcal{L}^v(\Phi(t))$. Then we have:

$$\mathcal{G}(\Phi(t+1)) \leq \mathcal{G}(\Phi(t)) - (\nu\eta_t - \frac{L_a\eta_t^2\beta^2}{2})\|\nabla\mathcal{L}^a(\Phi(t))\|^2 \tag{40}$$

By setting $\eta_t^a = \frac{\nu}{L_a\beta^2}$, we have:

$$\mathcal{G}(\Phi(t+1)) \leq \mathcal{G}(\Phi(t)) - \frac{\nu^2}{2L_a\beta^2}\|\nabla\mathcal{L}^a(\Phi(t))\|^2 \tag{41}$$

By using lemma (1), we have:

$$\mathcal{G}(\Phi(t+1)) \leq \mathcal{G}(\Phi(t)) - \frac{\nu^2\kappa^2}{2L_a\beta^2}|\mathcal{G}(\Phi(t))|^2. \tag{42}$$

Then we set $d = \frac{\nu^2\kappa^2}{2L_a\beta^2}$:

$$\frac{1}{\mathcal{G}(\Phi(t+1))} \geq \frac{1}{\mathcal{G}(\Phi(t)) - d|\mathcal{G}(\Phi(t))|^2} \tag{43}$$

$$\geq \frac{1}{\mathcal{G}(\Phi(t))}(1 + d\mathcal{G}(\Phi(t))) \tag{44}$$

$$= \frac{1}{\mathcal{G}(\Phi(t))} + d \tag{45}$$

By summing up for $k = 1, \cdots, T$, we have:

$$\mathcal{G}(\Phi(T)) \leq \frac{\mathcal{G}(\Phi(0))}{1 + dT\mathcal{G}(\Phi(0))} = \mathcal{O}(\frac{1}{T}). \tag{46}$$

# B  Notation Definition

We summarize the notation definition we used in this paper in Table I.

# C  Additional Experiments

## C.1  Datasets

**CREMAD:** CREMAD [4] is an audio-visual dataset designed for speech emotion recognition, It comprises 7,442 video clips of 2~3 seconds from 91 actors speaking several short words. which are divided into 6,698 training samples and 744 testing samples. This dataset includes the six most common emotions: angry, happy, sad, neutral, discarding, disgust and fear.

**KSounds:** KSounds [2] dataset is a commonly used dataset containing 31 action categories that can be recognized visually and auditorily, which contains 19,000 10~second video clips from YouTube. This dataset is divided into training set with 15,000 clips, validation set with 1,900 clips, and testing set with 1,900 clips.

**NVGesture:** NVGesture [31] dataset is a multimodal dataset specifically designed for gesture recognition, containing three types of data modalities, i.e., RGB, optical flow (OF), and Depth. This dataset includes 25 different gesture categories, covering a variety of common gestures. This dataset is divided into 1,050 samples for training and 482 samples for testing.

**VGGSound:** VGGSound [6] dataset is an audio-visual dataset in the wild, with nearly 200K 10-second video clips. Each sound-emitting object is also visible in the corresponding video clip in this dataset. After filtering out unavailable videos, 168,618 videos for training and validation, and 13,954 videos for testing in experimental settings.

**Twitter:** Twitter [51] dataset is a dataset designed for multilingual sentiment analysis, which is divided into training set with 3,197 pairs, validation set with 1,122 pairs and testing set with 1,037

Table I: Notation Definition

| Notation | Description |
|---|---|
| $N$ | The number of training data. |
| $K$ | The number of category labels. |
| $\boldsymbol{x}^a/\boldsymbol{x}^v$ | Audio/video data point. |
| $\boldsymbol{X} = \{(\boldsymbol{x}_i^a, \boldsymbol{x}_i^v)\}_{i=1}^N$ | Training set. |
| $\boldsymbol{y}_i \in \{0,1\}^K$ | Category label of $i$-th data. |
| $\boldsymbol{Y} = \{\boldsymbol{y}_i\}_{i=1}^N$ | Category label set. |
| Superscript $o$ | $o$-modality. |
| Superscript $a/v$ | Audio/video modality. |
| $\phi^o(\cdot)$ | Encoder of $o$ modality. |
| $\theta^o$ | Parameters of $\phi^o(\cdot)$. |
| $\boldsymbol{u}^o$ | Feature of $\boldsymbol{x}^o$. |
| $\psi^o(\cdot)$ | Classifier of $o$-modality. |
| $\Theta^o$ | Parameters of $\psi^o(\cdot)$. |
| $\boldsymbol{p}^o$ | Prediction of $\boldsymbol{x}^o$. |
| $\Phi^o = \{\theta^o, \Theta^o\}$ | Parameters set. |
| $\ell(\cdot)$ | Cross entropy loss. |
| $\mathcal{L}_{\text{CE}}(\cdot)$ | Cross entropy loss over training. |
| $\lambda$ | Hyper-parameter used to soften label. |
| $\hat{\boldsymbol{y}}_{it}^o$ | Residual label of $\boldsymbol{x}_i^o$ at $t$-th iteration. |
| $\epsilon(\cdot)$ | Classification loss. |
| $\epsilon_{\text{all}}(\cdot)$ | Loss for $t$ classifiers. |
| $\epsilon_{\text{pre}}(\cdot)$ | Loss for $t-1$ classifiers. |
| $L(\cdot)$ | Objective function. |
| $\mathcal{L}_{\text{SUB}}(\cdot)$ | Overall objective function. |
| $n^o$ | The number of classifier for $o$-modality. |
| $s_t^o$ | Confidence for $o$-modality at $t$-th iteration. |
| $\sigma$ | Coefficient of confidence score. |
| $\tau$ | Tolerance score. |
| $t_N$ | Adjustment frequency. |
| $\mathcal{G}(\cdot)$ | Gap function. |

pairs. This dataset consists of tweets collected from Twitter[2] with three different sentiment labels: positive, negative, and neutral.

**Sarcasm:** Sarcasm [3] dataset is specifically designed for sarcasm detection tasks, containing a large amount of text data labeled as sarcastic or non-sarcastic, which typically draws from various sources, such as social media, news comments, and conversation data, with each entry clearly labeled for sarcasm. This dataset includes 19,816 pairs for the training set, 2,410 pairs for the validation set, and 2,409 pairs for the testing set.

### C.2 Implementations of Toy Experiments

The toy experiment is designed to validate the motivation behind our method. Specifically, it is conducted on the CREMAD dataset, where the task is multimodal classification. Four methods are adopted for comparison: Naive MML, G-Blend [10], MML w/ GB, and Ours. Here, MML w/ GB refers to applying the gradient boosting algorithm to further improve the trained video model obtained from Naive MML, while keeping the audio model fixed. Ours refers to the method employing the sustained boosting algorithm proposed in this paper. Furthermore, ResNet-18 is used as the encoder for all methods, with its parameters randomly initialized. The selection of other hyperparameters is kept consistent with the main experiment.

---

[2]http://twitter.com

## C.3 More Results for Ablation Study

**MAP Results on CREMAD and KSounds Datasets:** We have added the MAP results of the ablation study on the CREMAD and KSounds datasets in Table II. The MAP results in Table II demonstrate that the objectives defined in Equation (2), (3), and (4) all lead to performance gains.

Table II: The MAP for ablation study on CREMAD and KSounds datasets.

| $\epsilon$ | $\epsilon_o$ | $\epsilon_p$ | CREMAD | | | KSounds | | |
|---|---|---|---|---|---|---|---|---|
| | | | Multi | Audio | Video | Multi | Audio | Video |
| ✓ | ✗ | ✓ | 0.8967 | 0.7061 | 0.7523 | 0.7796 | 0.5465 | 0.5484 |
| ✗ | ✓ | ✗ | 0.8895 | 0.7235 | 0.7535 | 0.7864 | **0.5482** | 0.5606 |
| ✗ | ✓ | ✓ | 0.9014 | 0.7321 | 0.7335 | 0.7821 | 0.5474 | 0.5651 |
| ✓ | ✓ | ✓ | **0.9103** | **0.7529** | **0.7612** | **0.7901** | 0.5461 | **0.6040** |

**Ablation Study on Image-Text and Tri-Modal Datasets:** To further validate our method, we report an ablation study on image-text dataset, i.e., Twitter dataset, and the dataset with three modalities, i.e., NVGesture dataset. The accuracy and F1 are reported in Table III and Table IV. The results demonstrate that the objectives defined in Equation (2), (3), and (4) all lead to performance gains in terms of both accuracy and F1.

Table III: The results for ablation study on Twitter dataset.

| $\epsilon$ | $\epsilon_o$ | $\epsilon_p$ | Accuracy | | | F1 | | |
|---|---|---|---|---|---|---|---|---|
| | | | Multi | Image | Text | Multi | Image | Text |
| ✓ | ✗ | ✓ | 0.7425 | 0.5400 | 0.7406 | 0.6830 | 0.3957 | 0.6868 |
| ✗ | ✓ | ✗ | 0.7396 | 0.5429 | 0.7387 | 0.6787 | **0.4113** | 0.6806 |
| ✗ | ✓ | ✓ | 0.7445 | 0.5265 | 0.7454 | 0.6767 | 0.3865 | 0.6794 |
| ✓ | ✓ | ✓ | **0.7512** | **0.5434** | **0.7488** | **0.6962** | 0.3998 | **0.6911** |

Table IV: The results for ablation study on NVGesture dataset.

| $\epsilon$ | $\epsilon_o$ | $\epsilon_p$ | Accuracy | | | | F1 | | | |
|---|---|---|---|---|---|---|---|---|---|---|
| | | | Multi | RGB | OF | Depth | Multi | RGB | OF | Depth |
| ✓ | ✗ | ✓ | 0.8465 | 0.7573 | 0.7780 | 0.7946 | 0.8505 | 0.7649 | 0.7809 | 0.7968 |
| ✗ | ✓ | ✗ | 0.8402 | 0.7427 | 0.7718 | 0.7759 | 0.8428 | 0.7604 | 0.7720 | 0.7812 |
| ✗ | ✓ | ✓ | 0.8485 | **0.8008** | 0.7988 | 0.7801 | 0.8526 | **0.8085** | 0.8009 | 0.7881 |
| ✓ | ✓ | ✓ | **0.8501** | 0.7988 | **0.8029** | **0.7967** | **0.8533** | 0.8076 | **0.8057** | **0.8033** |

## C.4 Evaluation of Generalization in More Scenarios

To further validate the effectiveness of our method, we employ a actor-independent data partitioning strategy on the CREMAD dataset, ensuring that the training, validation, and test sets contain mutually exclusive subjects following the setting of [18, 27]. Specifically, we randomly select the audio-video data of 63 actors for training, 14 actors for validation, and 14 actors for testing, with no overlap of actors among the three splits.

We first investigate the influence of the hyperparameters $\sigma$ and $\lambda$ on actor-independent CREMAD dataset. The results are shown in Figure I and Figure II. The results demonstrate that our method is not sensitive to hyper-parameters $\sigma$ and $\lambda$. Furthermore, the ablation study on actor-independent CREMAD dataset is shown in Table V. The results further verify the effectiveness of the objectives defined in Equation (2), (3), and (4).

Then, we compare our method with some representative methods on actor-independent CREMAD dataset. The results are shown in Table VI. From the results in Table VI, we can find that our method can achieve the best performance compared with SOTA baselines.

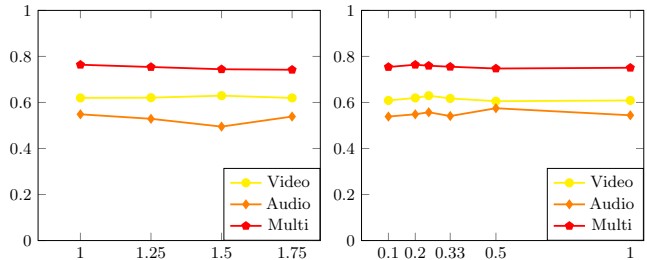

Figure I: Sensitivity to $\sigma$.     Figure II: Sensitivity to $\lambda$.

Table V: The results for ablation study on actor-independent CREMAD dataset.

| $\epsilon$ | $\epsilon_o$ | $\epsilon_p$ | Accuracy | | | MAP | | |
|---|---|---|---|---|---|---|---|---|
| | | | Multi | Image | Text | Multi | Image | Text |
| ✓ | ✗ | ✓ | 0.7443 | 0.6197 | 0.5191 | 0.8086 | 0.6517 | 0.5433 |
| ✗ | ✓ | ✗ | 0.7333 | 0.6262 | 0.5344 | 0.8135 | 0.6508 | **0.5861** |
| ✗ | ✓ | ✓ | 0.7388 | **0.6273** | 0.5366 | 0.8111 | 0.6496 | 0.5860 |
| ✓ | ✓ | ✓ | **0.7639** | 0.6087 | **0.5388** | **0.8300** | **0.6557** | 0.5544 |

Table VI: Comparison with SOTA baselines on actor-independent CREMAD dataset.

| Method | Accuracy | MAP |
|---|---|---|
| Audio/Video | 0.6219/0.5366 | 0.6588/0.5380 |
| Naive MML | 0.6153 | 0.6671 |
| MLA | 0.7421 | 0.8100 |
| ReconBoost | 0.7344 | 0.8027 |
| Ours | **0.7639** | **0.8300** |

## C.5 Influence of Unstable Confidence Score

Since our method relies on confidence gap to determine whether to add a classifier, fluctuations in confidence scores are indeed an influential factor. We conduct an experiment to explore the fluctuation of confidence score in scenario with modality noise on CREMAD dataset. Specifically, we add Gaussian noise to the video modality to introduce instability in confidence scores and investigate its impact on the algorithm. We design a metric to evaluate the fluctuation of confidence score during training. We first compute the absolute difference in confidence scores between two consecutive iterations, and then take the average over all training rounds:

$$\forall o \in \{a, v\}, \ \bar{s}^o = \frac{1}{n} \sum_{t=2}^{n} |s_t^o - s_{t-1}^o|. \tag{47}$$

Here, $s_t^o$ denotes the confidence score of $o$-modality at $t$-th iteration, and $n$ denotes the current iteration index in the training process. A higher value of $\bar{s}^o$ indicates more severe fluctuations in confidence scores, reflecting a sharper and less stable prediction behavior. Furthermore, we compare our method with naive MML and ReconBoost [21].

We first compare the model performance in scenario with modality noise in Table VII, where $\zeta$ denotes the noise rate, "#CLS" denotes the number of classifier during training, and in the first column, we also report the fluctuation of confidence score. In fact, as training progressed, we observe that in noisy scenarios, increasing the number of classifiers beyond a certain point no longer led to improved performance on the validation set. This phenomenon became more pronounced as the noise level increased. In other words, the classifiers may overcompensate in scenarios involving modality noise. On the other hand, we also observe that after reaching its optimal performance, the model's performance gradually degraded without a drastic drop. Therefore, in practical scenarios, an early stopping strategy based on validation set performance can be employed to select the optimal model and avoid classifier overcompensation.

Table VII: The fluctuation of confidence score.

| #CLS | $\zeta = 0, (\bar{s}^v = 0.1428)$ | $\zeta = 20, (\bar{s}^v = 0.3920)$ | $\zeta = 50, (\bar{s}^v = 0.4760)$ |
|---|---|---|---|
| 1 | 0.1411 | 0.1411 | 0.2863 |
| 2 | 0.6559 | 0.6129 | 0.6398 |
| 3 | 0.7634 | 0.7608 | 0.7728 |
| 4 | 0.8038 | 0.7917 | 0.8185 |
| 5 | 0.8266 | 0.8118 | 0.8212 |
| 6 | 0.8306 | 0.8293 | **0.8253** |
| 7 | 0.8401 | **0.8333** | 0.8199 |
| 8 | 0.8441 | 0.8266 | 0.8239 |
| 9 | 0.8468 | 0.8293 | 0.8185 |
| 10 | **0.8515** | 0.8280 | 0.8145 |

Furthermore, we compare the performance with competitive baselines including naive MML, ReconBoost [21]. The results are shown in Table VIII. We can find that compared with the competitive method ReconBoost, our approach consistently achieves superior performance, demonstrating its effectiveness in scenario with modality noise.

Table VIII: Performance comparison under modality noise scenario.

| Method | $\zeta = 0$ | $\zeta = 20$ | $\zeta = 50$ |
|---|---|---|---|
| Naive MML | 0.6507 | 0.6425 | 0.6331 |
| ReconBoost | 0.7557 | 0.7215 | 0.7031 |
| Ours | **0.8515** | **0.8333** | **0.8253** |

## C.6 Computational and Memory Cost at Inference Stage

To demonstrate the practical applicability of our approach, we further provide a comprehensive analysis of its computational and memory overhead at inference stage.[3] Furthermore, we conduct an experiment to analyze the computational and memory cost during inference phase on CREMAD dataset. The baselines include naive MML, PMR [12], AGM [28], MLA [52], ReconBoost [21]. The results are shown in Table IX. We can find that our method achieves superior performance while maintaining competitive inference time. Furthermore, compared with naive MML, our method introduces only 1M additional parameters when 10 classifiers are added. Given the total model size of 23.6M, this corresponds to a relatively small increase of approximately 4%.

Table IX: Computational cost comparison on CREMAD dataset.

| Method | Accuracy | Inference time (s) |
|---|---|---|
| Naive MML | 0.6507 | 5.29 |
| PMR | 0.6659 | 6.59 |
| AGM | 0.6733 | 5.58 |
| MLA | 0.7943 | 5.63 |
| ReconBoost | 0.7557 | 5.33 |
| Ours | 0.8515 | 5.62 |

## C.7 Selection of Score Function

In our method, the score function is employed to quantify the disparity in classification performance between different modalities, which serves as the basis for determining whether to introduce an additional classifier for the weak modality. Therefore, any metric that effectively captures the

---

[3]For computational cost during training, the analysis have been posted in Section 4.5 of the original paper.

difference in classification capabilities across modality-specific models can be adopted as a score function.

To some extent, entropy and loss can reflect the learning status of modality-specific models and thereby serve as proxies for their classification capabilities. To evaluate the effectiveness of using entropy and loss as score functions, we conducted experiments on the CREMAD dataset, where these two metrics were used to guide the adaptive classifier assignment. The experimental results are presented in Table X, where $\tau$ denotes the threshold. Please note that since we use the ratio between different modal metrics to compare against the threshold $\tau$, the value of $\tau$ remains consistent across different metric types such as confidence, loss, and entropy. These findings indicate that the proposed method is robust to the choice of score function, consistently achieving comparable results regardless of whether confidence score, entropy, or loss is used.

Table X: Performance with different score function.

| Score function | Accuracy | Threshold $\tau$ | #CLS |
|---|---|---|---|
| Confidence score (Ours) | 0.8515 | 0.01 | 10 |
| Entropy | 0.8522 | 0.01 | 10 |
| Loss | 0.8562 | 0.01 | 10 |

## C.8 Computational Cost vs. The Number of Classifiers

The computational overhead—including both training and inference time—is indeed influenced by the number of classifiers. Since our method dynamically adds classifiers during training, we incorporate a threshold to limit the maximum number of classifiers. This allows us to systematically investigate the trade-off between computational cost and model performance as a function of classifier quantity.

Specifically, we conduct experiments on the CREMAD dataset, where the maximum number of weak modality classifiers is constrained to not exceed a predefined threshold $M$. We vary $M$ across the set $\{2, 4, 6, 8, 10\}$, and for each setting, we report the training time, inference time, and the corresponding classification performance.

The experimental results are summarized in Table XI. We observe that as the number of classifiers increases, both training time and inference time increase accordingly, while performance also improves to a certain extent. This indicates that incorporating more classifiers can indeed enhance model performance, but at the cost of increased computational overhead.

Table XI: Computational cost vs the number of classifiers.

| #CLS | Accuracy | Training time (hrs) | Inference time (s) |
|---|---|---|---|
| 2 | 0.8159 | 1.68 | 5.31 |
| 4 | 0.8387 | 1.72 | 5.38 |
| 6 | 0.8441 | 1.78 | 5.46 |
| 8 | 0.8468 | 1.86 | 5.53 |
| 10 | 0.8515 | 1.98 | 5.62 |

