# OpenReview forum: "Rethinking Multimodal Learning from the Perspective of Mitigating Classification Ability Disproportion"
_NeurIPS.cc/2025/Conference — NeurIPS 2025 oral_

### Official Review · Reviewer_3wbR · 2025-06-29

**Clarity:** 4
**Significance:** 3
**Originality:** 4
**Rating:** 5
**Confidence:** 5

**Summary:**

This paper tackles the persistent challenge of modality imbalance in multimodal learning, proposing that the core issue stems from the inherent classification ability gap between modalities. The authors introduce a sustained boosting approach that incrementally strengthens weaker modality classifiers via residual error minimization. Additionally, an adaptive strategy dynamically assigns new classifiers based on modality-specific confidence scores. The theoretical convergence analysis is solid and bolstered by empirical results showing superior performance across diverse benchmarks. The results of visualizations, t-SNE plots, and convergence curves provides transparency into model behavior. However, despite the impressive results, the paper could benefit from further analysis on efficiency and more diverse ablation studies. Overall, this paper contributes a fresh viewpoint and a robust solution to the community.

**Questions:**

My key concerns about the paper are outlined below:

(1). The relationship between training cost and the number of classifiers, as well as the corresponding performance, warrants further discussion.

(2) The authors are encouraged to report the results of ablation studies on additional datasets to validate the generality of the proposed method.

**Ethical Concerns:**

["NO or VERY MINOR ethics concerns only"]

**Final Justification:**

The proposed method is novel, its convergence is theoretically guaranteed. During the rebuttle, the ahthous have well responsed my concerns. Besides, all reviewers consistently tends to accept it. So, I finnally give it 5:Accept.

**Limitations:**

Yes

**Paper Formatting Concerns:**

Not formatting concerns.

**Quality:**

3

**Strengths And Weaknesses:**

This paper proposes a novel method to address the multimodal imbalance problem, with the following key advantages:
(1)  the paper addresses a real and understudied issue in multimodal learning, i.e., disproportionate classification ability across modalities. It further identifies this problem as a key contributor to modality imbalance. Furthermore, it introduces a novel sustained boosting algorithm that explicitly enhances weaker modality classifiers;
(2) theoretical analysis (Theorem 1) supports convergence guarantees of the proposed gap minimization framework;
(3) consistent gains across metrics (Acc/MAP/F1) and modalities (video, audio, text) in Table 1 are impressive. Code release and exhaustive implementation details (Sec 4.1) align with open-science best practices;
(4) the paper is clearly written, logically coherent, and well organized, making it easy to follow and understand.

Furthermore, several weaknesses in the paper should be noted: (1). Lemma 1 only provides an abbreviated title, and it appears that the full statement or introduction may be missing; (2). Details on computational cost vs. performance tradeoff are limited and deserve deeper discussion. Training overhead (Fig 5) grows linearly with classifiers. This is problematic for resource-limited settings, yet mitigation strategies are absent; (3). The ablation study is primarily conducted on one dataset (CREMAD), which may not generalize well.

---

> ### Author Rebuttal · Authors · 2025-07-30
>
> Thank you for your conductive comments.
>
> **Answer for Question "Lemma 1 Statement"**: Thank you for pointing this out. This was indeed a typographical error on our part. The corrected statement of the lemma is as follows:\
> **Lemma 1 (Gap Bound)** There exists a constant $\kappa$ such that the loss gap function $\mathcal{G}(\cdot)$ and the gradient norm of the strong modality satisfy the following relationship:
>
> $$||\nabla\mathcal{L}^a(\Phi_t)||\ge\kappa|\mathcal{G}(\Phi_t)|.$$
>
> We will address this issue in the final version.
>
> **Answer for Question: "Computational Cost vs The Number of Classifiers"**: The computational overhead—including both training and inference time—is indeed influenced by the number of classifiers. Since our method dynamically adds classifiers during training, we incorporate a threshold to limit the maximum number of classifiers. This allows us to systematically investigate the trade-off between computational cost and model performance as a function of classifier quantity.
>
> Specifically, we conduct experiments on the CREMA-D dataset, where the maximum number of weak modality classifiers is constrained to not exceed a predefined threshold $M$. We vary $M$ across the set {2, 4, 6, 8, 10}, and for each setting, we report the training time, inference time, and the corresponding classification performance.
>
> The experimental results are summarized in the following Table. We observe that as the number of classifiers increases, both training time and inference time increase accordingly, while performance also improves to a certain extent. This indicates that incorporating more classifiers can indeed enhance model performance, but at the cost of increased computational overhead.
>
> *Table 1. Computational cost vs the number of classifiers.*
>
> | #CLS         | Accuracy | Training time (hrs) | Inference time (s) |
> |:----:|:----:|:----------:|:------------:|
> | 2            | 0.8159   | 1.68                | 5.31               |
> | 4            | 0.8387   | 1.72                | 5.38               |
> | 6            | 0.8441   | 1.78                | 5.46               |
> | 8            | 0.8468   | 1.86                | 5.53               |
> | 10           | 0.8515   | 1.96                | 5.62               |
>
> Relevant experiments and discussion will be included in the final version.
>
> **Answer for Question: "Ablation Study on Other Dataset"**: To further demonstrate the effectiveness of the key components of our method, we conduct the experiments for ablation study on trimodal dataset, i.e., NVGeasture dataset. Specifically, we analyze the effectiveness of the losses $\epsilon$, $\epsilon_o$, and $\epsilon_p$ respectively defined in Eq.(2), (3), and (4) of the original paper.
>
> For video-audio, image-text, and tri-modal scenarios, we selected three representative datasets for ablation experiments: the video-audio dataset CREMA-D, the image-text dataset Twitter2015, and the tri-modal dataset NVGesture. The result on CREMA-D dataset is presented in Table 2 of the original paper. And the results on Twitter2015 and NVGeasture dataset are respectively shown in Table 2 and Table 3. We can find that both objectives in Eq. (2), (3), and (4) can boost the multimodal classification performance in terms of accuracy.
>
> *Table 2. Ablation study on Twitter2015 dataset.*
> | $\epsilon$ | $\epsilon_o$ | $\epsilon_p$ | Multi  | Image  | Text   |
> |:----:|:----:|:----------:|:------------:|:------------:|:------------:|
> |  yes       | no    | yes   | 0.7425 | 0.5400 | 0.7406 |
> |  no        | yes   | no    | 0.7396 | 0.5429     | 0.7387 |
> |  no        | yes   | yes   | 0.7445 | 0.5265 | 0.7454 |
> |  yes       | yes   | yes   | **0.7512** | **0.5434** | **0.7488** |
>
> *Table 3. Ablation study on NVGeasture dataset.*
>
> | $\epsilon$ | $\epsilon_o$ | $\epsilon_p$  | Multi | RGB  | OF |  Depth |
> |:----:|:----:|:----------:|:------------:|:------------:|:------------:|:------------:|
> |  yes       | no    | yes        | 0.8465 |0.7573        |0.7780        |0.7946        |
> |  no        | yes   | no         | 0.8402 |0.7427        |0.7718        |0.7759        |
> |  no        | yes   | yes        | 0.8485 |**0.8008**        |0.7988        |0.7801        |
> |  yes       | yes   | yes        | **0.8501** |0.7988        |**0.8029**        |**0.7967**        |
>
>
> Relevant experiments and discussion will be included in the final version.

---

> > ### Comment · Reviewer_3wbR · 2025-08-01
> >
> > The authors have well responsed my concerns, I keep my positive score.

---

### Official Review · Reviewer_h6gC · 2025-06-29

**Clarity:** 4
**Significance:** 4
**Originality:** 3
**Rating:** 5
**Confidence:** 4

**Summary:**

This paper proposes a promising and adaptable framework for improving multimodal learning in scenarios where certain modalities dominate others in classification performance. The proposed sustained boosting and adaptive classifier assignment modules offer an elegant way to dynamically rebalance classification capacity without heavily redesigning the architecture. Results show strong empirical gains across six datasets. However, a more thorough comparison and discussion of the similarities and differences with existing boosting-based method, i.e., ReconBoost, are necessary to better highlight the novelty of the proposed approach. Nevertheless, this is a meaningful and actionable contribution to the field of robust multimodal systems.

**Questions:**

1. In what ways is the proposed method innovative compared to ReconBoost?

2. No ablation on the confidence scoring function—would other metrics (entropy, loss) work better?

3. Although the authors demonstrate the effectiveness of their method on a trimodal dataset, relevant experimental details are lacking.

**Ethical Concerns:**

["NO or VERY MINOR ethics concerns only"]

**Final Justification:**

My concerns are addressed, and I will keep my positive rating.

**Limitations:**

yes

**Paper Formatting Concerns:**

No Paper Formatting Concerns

**Quality:**

4

**Strengths And Weaknesses:**

## Strengths:
1. Novelty of the Proposed Method: The proposed multimodal learning approach to address the modality imbalance issue is applicable to real-world multimodal systems where sensor strengths vary. And the adaptive classifier assignment strategy reduces the need for manual model tuning during training.

2. Flexibility of the Proposed Method: Successful extension to 3 modalities (NVGesture) proves versatility beyond audio-visual/image-text tasks. Performance gains are consistent across tasks with more than two modalities, showing versatility.

3. Effectiveness of the Experiments: Extensive experimental results demonstrate the effectiveness of the proposed method. Experiments include meaningful evaluation metrics (accuracy, MAP, MacroF1) suitable for practical deployment.

4. Theoretical Proof: The convergence guarantees presented in Section 3.4 reinforce the credibility of the empirical results.

## Weaknesses:
1. Comparison with ReconBoost [1]: The authors propose a boosting algorithm to address the problem of modality imbalance. A detailed comparison with the ReconBoost method, including both similarities and differences, is necessary to clarify the contributions of the proposed approach.

2. Selection of Scoring Function: The author employs the existing scoring method OGM [2]; however, it would be valuable to discuss whether more effective alternatives are available.

Reference:
[1]. Hua, Cong, et al. Reconboost: Boosting can achieve modality reconcilement. ICML, 2024.
[2]. Peng, Xiaokang, et al. Balanced multimodal learning via on-the-fly gradient modulation. CVPR, 2022.

---

> ### Author Rebuttal · Authors · 2025-07-30
>
> Thank you for your conductive comments.
>
> **Answer for Question "Comparison with ReconBoost"**: Although both our method and ReconBoost [1] are inspired by the idea of gradient boosting, they differ fundamentally in motivation, implementation, and theoretical grounding:
> - The two methods are driven by distinct objectives. Our approach aims to improve the classification performance of weak modality classifiers using gradient boosting, whereas ReconBoost adopts the concept of gradient boosting to mitigate modality competition.
> - The utilization of gradient boosting differs significantly between these two methods. Our method learns the residual of a single-modality classifier to refine weak modality performance, while ReconBoost leverages the residual from another modality to address inter-modal competition.
> - The theoretical foundations of the two methods diverge. Our method establishes a link between the gradient boosting algorithm and a performance gap function, whereas ReconBoost demonstrates that, under certain conditions, introducing KL divergence as a regularization term results in a training objective equivalent to a specific form of gradient boosting.
>
> In summary, despite both methods drawing inspiration from gradient boosting, they embody fundamentally different motivations, mechanisms, and theoretical interpretations.
>
> Relevant discussion will be included in the final version.
>
> **Answer for Question "Selection of Score Function"**: In our method, the score function is employed to quantify the disparity in classification performance between different modalities, which serves as the basis for determining whether to introduce an additional classifier for the weak modality. Therefore, any metric that effectively captures the difference in classification capabilities across modality-specific models can be adopted as a score function.
>
> To some extent, entropy and loss can reflect the learning status of modality-specific models and thereby serve as proxies for their classification capabilities. To evaluate the effectiveness of using entropy and loss as score functions, we conducted experiments on the CREMA-D dataset, where these two metrics were used to guide the adaptive classifier assignment. The experimental results are presented in Table 1, where $\tau$ denotes the threshold. Please note that since we use the ratio between different modal metrics to compare against the threshold $\tau$, the value of $\tau$ remains consistent across different metric types such as confidence, loss, and entropy. These findings indicate that the proposed method is robust to the choice of score function, consistently achieving comparable results regardless of whether confidence score, entropy, or loss is used.
>
> *Table 1. Performance with different score function.*
>
> | Score function | Accuracy | Threshold $\tau$ | #CLS |
> |:----:|:----:|:----------:|:------------:|
> | Confidence score (Ours)  | 0.8515 | 0.01    | 10   |
> | Entropy        | 0.8522  | 0.01             | 10   |
> | Loss           | 0.8562  | 0.01             | 10   |
>
> Relevant experiments and discussion will be included in the final version.
>
> **Answer for Question "Trimodal Experimental Details"**: For the scenario with three modalities, a key aspect of the experiment lies in determining whether to add an additional classifier based on the confidence scores of the three modalities. Specifically, we compare the confidence score of each modality with the average of the confidence scores of the other two. If the difference exceeds a predefined threshold, a new classifier is assigned to the corresponding modality.
>
> Relevant implementation details will be included in the final version.
>
> **Reference:**\
> [1]. Hua, Cong, et al. ReconBoost: Boosting can achieve modality reconcilement. *ICML*. 2024.

---

> > ### Comment · Reviewer_h6gC · 2025-08-08
> >
> > Thanks for the response! My concerns are addressed, and I will keep my positive rating.

---

### Official Review · Reviewer_5gtj · 2025-06-30

**Clarity:** 3
**Significance:** 3
**Originality:** 3
**Rating:** 5
**Confidence:** 5

**Summary:**

This manuscript introduces a theoretically sound and empirically validated approach to address modality imbalance in multimodal learning. By crafting a specialized boosting algorithm, it enhances the classification ability of weaker modality-specific classifiers. Through an analysis of multimodal model capacity, the work sheds fresh light on the roots of modality imbalance. Comprehensive evaluations against state-of-the-art methods demonstrate the proposed approach’s clear advantages, and supplementary experiments further attest to its overall efficacy. However, the paper would benefit from deeper discussion and additional experiments regarding the selection of the boosting strategy and the broader applicability of the technique.

**Questions:**

1. Could other boosting paradigms yield comparable improvements, and what underlying mechanisms would support their success?
2. In what ways could this boosting-based approach be adapted for or integrated into early fusion multimodal frameworks?

**Ethical Concerns:**

["NO or VERY MINOR ethics concerns only"]

**Final Justification:**

I appreciate the author's response, which provided ample clarification and addressed my concerns. Therefore, I'm willing to maintain my rating.

**Limitations:**

yes

**Quality:**

4

**Strengths And Weaknesses:**

**Strengths:**

1. The authors reconceptualize modality imbalance as a disparity in classification capacities, an angle largely overlooked in previous studies, which prompts a reconsideration of the problem through the lens of model capacity.
2. The boosting framework is effectively tailored for the multimodal context, marking a creative application of boosting techniques.
3. Experiments conducted on six diverse datasets consistently show superior performance compared to leading baselines such as ReconBoost and DI‑MML.
4. The work includes extensive analyses—hyperparameter sweeps, ablation trials, and visualizations—to underscore the proposed method’s strengths.
5. Figure 4 clearly shows improved classification outcomes for weaker modalities, bolstering confidence in the approach.

**Weaknesses:**

1. Since the boosting algorithm is rooted in gradient boosting, it remains uncertain whether other boosting variants could match its performance.
2. The current method is validated only for late fusion; its effectiveness with early fusion architectures is not explored.
3. The annotation “Full Prediction vs. t−1 CLS” in Figure 3 is unclear—more precise labeling or explanation is needed.
4. The design and parameters of the toy experiments are insufficiently described; supplying full experimental details would enhance reproducibility.

---

> ### Author Rebuttal · Authors · 2025-07-30
>
> Thank you for your conductive comments.
>
> **Answer for Question "Adaptability for Other Boosting Strategy"**: First, our algorithm is inspired by the gradient boosting paradigm. The motivation behind this approach lies in improving the classification performance of weak modalities during training, for which residual learning provides a natural solution. Second, other representative methods such as AdaBoost also enhance classifier performance by modifying the sampling strategy, which is another feasible direction. We believe that resampling misclassified samples from the weak modality can facilitate more effective learning, potentially bridging our motivation with AdaBoost-style algorithms.
>
> It is worth noting that certain existing works have investigated sample-level imbalance. For example, SMV [1] achieves balanced learning by resampling modality-specific data that has not been sufficiently learned, and has demonstrated promising results. However, integrating this sample-level resampling strategy into the weak modality boosting framework may constitute a novel research direction. As such, we leave this exploration for future work.
>
> Relevant discussion will be included in the final version.
>
> **Answer for Question "Effectiveness with Early Fusion Architectures"**: Our method is built upon a late fusion architecture, with the core motivation of identifying and mitigating performance disparities across modalities, thereby achieving balanced classification capabilities among them. This implies that our method cannot be directly applied to architectures that rely solely on a single fusion classifier. In early fusion methods, only fused representations are used for prediction, and the classification process is not decomposable into modality-specific classifiers. As a result, our approach is not directly compatible with early fusion architectures.
>
> To adapt our method to early fusion settings, it may be necessary to explicitly decouple the classification contributions of individual modalities and assess their performance differences. This would likely require a fundamentally new architectural design, which we consider a promising direction for future work.
>
> **Answer for Question "Explation of Full Prediction vs $t-1$ CLS"**: Our method adaptively adds new classifiers during the training process, and we denote the current number of classifiers for the weak modality as $t$. "Prediction of $t-1$ CLS" refers to the performance of the first $t-1$ classifiers, excluding the newly added one, while "Full Prediction" refers to the performance of all t classifiers.
>
> The primary objective of this experiment is to verify the feasibility and necessity of residual learning by comparing the performance of the $t$-th classifier with that of the first $t–1$ classifiers. Essentially, the $t$-th classifier learns the residual—that is, the performance difference between itself and the previous classifiers. This observation is also supported by the results shown in Figure 3 of the original paper.
>
> **Answer for Question "Implementations of Toy Experiments"**: The toy experiment is designed to validate the motivation behind our method. Specifically, it is conducted on the CREMA-D dataset, where the task is multimodal classification. Four methods are adopted for comparison: *Naive MML*, *G-Blend* [2], *MML w/ GB*, and *Ours*. Here, MML w/ GB refers to applying the gradient boosting algorithm to further improve the trained video model obtained from Naive MML, while keeping the audio model fixed. Ours refers to the method employing the sustained boosting algorithm proposed in this paper. Furthermore, ResNet-18 is used as the encoder for all methods, with its parameters randomly initialized. The selection of other hyperparameters is kept consistent with the main experiment.
>
> Relevant implementation details will be included in the final version.
>
> **Reference:**\
> [1]. Wei, Yake, et al. Enhancing multimodal cooperation via sample-level modality valuation. *CVPR*. 2024.\
> [2]. Wang, Weiyao, Du Tran, and Matt Feiszli. What makes training multi-modal classification networks hard?. *CVPR*. 2020.

---

> > ### Comment · Reviewer_5gtj · 2025-08-04
> >
> > The authors have responded well to my concerns, and I keep my score.

---

### Official Review · Reviewer_Y2TP · 2025-07-05

**Clarity:** 3
**Significance:** 3
**Originality:** 3
**Rating:** 4
**Confidence:** 3

**Summary:**

This paper addresses the often-overlooked issue of modality imbalance in multimodal learning (MML), particularly the disproportion in classification ability between strong and weak modalities. Instead of aligning the learning rates or optimizing joint features, the authors focus on directly enhancing the classification performance of the weaker modality. To achieve this, they propose a Sustained Boosting algorithm, which jointly minimizes classification and residual errors to progressively improve weak classifiers. Additionally, an Adaptive Classifier Assignment (ACA) strategy is introduced to dynamically assign more classifiers to underperforming modalities, guided by confidence scores. Theoretical analysis is provided to guarantee the convergence of the proposed method in terms of closing the loss gap between modalities. Empirical results across six multimodal datasets—including audio-video and image-text pairs—demonstrate that the proposed approach consistently outperforms a wide range of state-of-the-art baselines, including ReconBoost, LFM, and others. Ablation and sensitivity analyses further validate the contribution of each component.

**Questions:**

Adaptive Classifier Assignment: If the confidence score for a modality fluctuates due to noise or temporary degradation in input quality, does the algorithm risk overcompensating by adding unnecessary classifiers, potentially leading to overfitting? It would also be helpful to see results from a stress test on this adaptive strategy under noisy modality conditions.

How does your method behave when a modality is missing or corrupted? Given that the method is trained on full-modal input, would sustained boosting and classifier assignment degrade gracefully if a modality is unavailable during inference?

A natural baseline to compare against would be modality-specific knowledge distillation, where weak modalities learn from stronger ones via teacher-student dynamics. Has this been explored or considered?

**Ethical Concerns:**

["NO or VERY MINOR ethics concerns only"]

**Limitations:**

See questions

**Quality:**

3

**Strengths And Weaknesses:**

The paper exhibits strong technical rigor and proposes a clearly motivated shift in focus—from aligning learning dynamics to balancing classification abilities across modalities. This perspective is refreshing and highlights a previously under-discussed source of performance degradation in MML. The use of a boosting-based framework is well-justified, and the idea of sustaining the optimization over both classification and residual errors is thoughtfully executed. Importantly, the inclusion of the Adaptive Classifier Assignment strategy provides a dynamic mechanism to address imbalance throughout training, rather than relying on fixed configurations.

Moreover, the authors support their claims with extensive experiments across six datasets that cover diverse modalities and tasks. Theoretical convergence analysis is presented with full proofs, ensuring the robustness of the proposed approach. The empirical results are generally strong, often yielding the best performance on each benchmark.

However, the work is not without its limitations. While the authors do touch on model overhead, the computational and memory cost introduced by using multiple classifiers—especially during inference—is not analyzed in detail. This raises concerns about the scalability and deployability of the approach in real-world applications, particularly those requiring low-latency or resource-constrained inference. Additionally, although the performance gain is consistent, the margin over strong baselines like ReconBoost or LFM is sometimes modest. Another limitation is that the approach is designed for cases where all modalities are fully present, and the framework’s behavior under partial or missing modality scenarios is not explored. Finally, while the adaptive strategy is a novel idea, there is a lack of discussion on its stability—e.g., how it reacts to noisy modality-specific confidence scores or abrupt changes in modality quality during training.

---

> ### Author Rebuttal · Authors · 2025-07-30
>
> Thank you for your conductive comments.
>
> **Answer for Question "Influence of Unstable Confidence Score"**: Since our method relies on confidence gap to determine whether to add a classifier, fluctuations in confidence scores are indeed an influential factor. We conduct an experiment to explore the fluctuation of confidence score in scenario with modality noise on CREMA-D dataset. Specifically, we add Gaussian noise to the video modality to introduce instability in confidence scores and investigate its impact on the algorithm. We design a metric to evaluate the fluctuation of confidence score during training. We first compute the absolute difference in confidence scores between two consecutive iterations, and then take the average over all training rounds, i.e., $\bar{s}=\frac{1}{n}\sum_{t=2}^n|s_t-s_{t-1}|$. Here, $s_t$ denotes the confidence score at $t$-th iteration. A higher value of $\bar{s}$ indicates more severe fluctuations in confidence scores, reflecting a sharper and less stable prediction behavior. Furthermore, we compare our method with naive MML and ReconBoost [1].
>
> We first compare the model performance in scenario with modality noise in Table 1, where $\mu$ denotes the noise rate [2], #CLS denotes the number of classifier during training, and in the first column, we also report the fluctuation of confidence score. In fact, as training progressed, we observe that in noisy scenarios, increasing the number of classifiers beyond a certain point no longer led to improved performance on the validation set. This phenomenon became more pronounced as the noise level increased. In other words, the classifiers *may overcompensate* in scenarios involving modality noise. On the other hand, we also observe that after reaching its optimal performance, the model's performance gradually degraded without a drastic drop. Therefore, in practical scenarios, an early stopping strategy based on validation set performance can be employed to select the optimal model and avoid classifier overcompensation.
>
> *Table 1. The fluctuation of confidence score.*
>
> | #CLS | &nbsp;&nbsp;&nbsp;&nbsp;$\mu=0\%$, ($\bar{s}=0.1428$)&nbsp;&nbsp;&nbsp;&nbsp; | &nbsp;&nbsp;&nbsp;&nbsp;$\mu=20\%$, ($\bar{s}=0.3920)$&nbsp;&nbsp;&nbsp;&nbsp; | &nbsp;&nbsp;&nbsp;&nbsp;$\mu=50\%$, ($\bar{s}=0.4760)$&nbsp;&nbsp;&nbsp;&nbsp; |
> |:----:|:-----------------------------:|:-----------------------------:|:-----------------------------:|
> | 1 | 0.1411                  | 0.1411 | 0.2863 |
> | 2 | 0.6559                  | 0.6129 | 0.6398 |
> | 3 | 0.7634                  | 0.7608 | 0.7728 |
> | 4 | 0.8038                  | 0.7917 | 0.8185 |
> | 5 | 0.8266                  | 0.8118 | 0.8212 |
> | 6 | 0.8306                  | 0.8293 | **0.8253** |
> | 7 | 0.8401                  | **0.8333** | 0.8199 |
> | 8 | 0.8441                  | 0.8266 | 0.8239 |
> | 9 | 0.8468                  | 0.8293 | 0.8185 |
> | 10 | **0.8515**              | 0.8280 | 0.8145 |
>
> Furthermore, we compare the performance with competitive baselines including naive MML, ReconBoost [1]. The results are shown in Table 2. We can find that compared with the competitive method ReconBoost, our approach consistently achieves superior performance, demonstrating its effectiveness in scenario with modality noise.
>
> *Table 2. Performance comparison under modality noise scenario.*
>
> | Method  | $\mu=0\%$| $\mu=20\%$ | $\mu=50\%$ |
> |:----:|:----:|:----------:|:------------:|
> | Naive MML  | 0.6507   | 0.6425     | 0.6331     |
> | ReconBoost | 0.7557   | 0.7215     | 0.7031     |
> | Ours       |**0.8515**|**0.8333**  |**0.8253**  |
>
> Relevant experiments and discussion will be included in the final version.
>
> **Answer for Question "Stability under Modality Missing"**: Our method is adaptable to scenarios involving missing modalities. We comprehensively evaluate its performance under test-time missing modality conditions. Specifically, test-time missing refers to cases where the modalities are complete during training but missing during the testing phase. The experiments are conducted on CREMA-D dataset with different missing rate [3]. Naive MML, ReconBoost [1], and MLA [3] are selected as baselines for comparison, where MLA introduces specifically designed algorithms to address the corresponding challenges in modality missing. We report the results with missing rate 20% and 50% in Table 3. We can find that as the modality missing rate increases, the performance of all methods declines. Nevertheless, our method consistently achieves the best performance under all missing rates, demonstrating the effectiveness of our method in scenario with missing modality.
>
> *Table 3. Performance comparison in scenario with modality missing.*
>
> | Method  | $r=0\%$  | $r=20\%$ | $r=50\%$ |
> |:----:|:----:|:----------:|:------------:|
> | Naive MML  | 0.6507   | 0.5849   | 0.5242   |
> | ReconBoost | 0.7557   | 0.6321   | 0.5568   |
> | MLA        | 0.7943   | 0.6935   | 0.5753   |
> | Ours       |**0.8515**|**0.7540**|**0.6008**|
>
> Relevant experiment and discussion will be included in the final version.
>
> **Answer for Question "Modality-Specific Knowledge Distillation Strategy"**: Using strong modalities to distill weaker ones is indeed a compelling and insightful idea. In our approach, we utilize performance gap to determine the relative strength of each modality and then independently enhance the classification capability of the weaker modalities. This can be viewed as an *implicit* strategy that leverages modality performance to boost the learning of weak modalities. In contrast, methods that *explicitly* distill knowledge from strong modalities aim to directly improve the classification performance of weaker ones through knowledge transfer. For instance, C2KD [4] is a representative method designed based on this idea by using the confidence scores from the strong modality to guide and refining the confidence of the weak modality. We conducted experiments to compare the effectiveness of our method and C2KD.
>
> We compare this approach with our method on CREMA-D dataset under the same setting. The accuracy results are reported in Table 5, where the accuracy of C2KD is refered from its original paper. We can find that (1). Compared with naive MML, C2KD can achieve better performance, demonstrating the effectiveness of modality-specific knowledge distillation; (2). Our approach can achieve the best performance compared with naive MML and C2KD. This suggests that directly enhancing the performance of weak modality classifiers may be a more effective strategy than knowledge distillation.
>
> *Table 5. The effectiveness of cross-modal knowledge distillation.*
>
> | Method     | Audio    | Video    | Multi    |
> |:----:|:----:|:----------:|:------------:|
> | Naive MML    | 0.6075   | 0.2702   | 0.6507   |
> | C2KD*        | 0.614    | 0.628    | -        |
> | Ours         |**0.6835**|**0.6828**|**0.8515**|
>
> Relevant experiments and discussion will be included in the final version.
>
> **Answer for Question "Computational and Memory Cost"**: To demonstrate the practical applicability of our approach, we further provide a comprehensive analysis of its computational and memory overhead. For computational cost during training, the analysis have been posted in Section 4.5 of the original paper. Furthermore, we conduct an experiment to analyze the computational and memory cost during inference phase on CREMA-D dataset. The baselines include naive MML, PMR, AGM, MLA, ReconBoost. The results are shown in Table 6. We can find that our method achieves superior performance while maintaining competitive inference time. Furthermore, compared with naive MML, our method introduces only 1M additional parameters when 10 classifiers are added. Given the total model size of 23.6M, this corresponds to a relatively small increase of approximately 4%.
>
> *Table 6. Computational cost comparison on CREMA-D dataset.*
>
> |  Method     | Accuracy | Inference time (s) |
> |:----:|:----:|:----------:|
> | Naive MML    | 0.6507   | 5.29               |
> | PMR          | 0.6659   | 6.59               |
> | AGM          | 0.6733   | 5.58               |
> | MLA          | 0.7943   | 5.63               |
> | ReconBoost   | 0.7557   | 5.33               |
> | Ours         | **0.8515**   | 5.62               |
>
> Relevant experiments and discussion will be included in the final version.
>
> **Answer for Question "Performance Comparison"**: From the Table 1 of the original paper, we can find that our method achieves notable performance improvements on both video-audio and tri-modal datasets. Specifically, it improves accuracy by 1.53% on the CREMA-D dataset, 0.10% on the KSounds dataset, 0.27% on VGG dataset, and 0.36% on the NVGesture dataset. However, on the Twitter2015 and Sarcasm datasets, the performance gains are relatively modest, although our method still achieves the best results in most cases. This may be attributed to the limited complementary information between modalities in the text and image data. In fact, we observe that, compared to the best single-modal performance on these datasets, the improvements brought by all multimodal methods are generally minor. Notably, on the Twitter2015 dataset, most multimodal approaches even underperform the best single-modal baseline. Addressing this question is a promising direction for future research.
>
> **Reference:**\
> [1]. Hua, Cong, et al. ReconBoost: Boosting can achieve modality reconcilement. *ICML*. 2024.\
> [2]. Hu, Peng, et al. Learning cross-modal retrieval with noisy labels. *CVPR*. 2021.\
> [3]. Zhang, Xiaohui, et al. Multimodal representation learning by alternating unimodal adaptation. *CVPR*. 2024.\
> [4]. Huo, Fushuo, et al. C2kd: Bridging the modality gap for cross-modal knowledge distillation. *CVPR*. 2024.

---

> > ### Author Response · Authors · 2025-08-07
> > **Kind Reminder to Reviewer Y2TP**
> >
> > Dear Reviewer,
> >
> > I hope this message finds you well. As the discussion period is nearing its end, with less than two days remaining, I want to ensure that we have addressed all of your concerns satisfactorily.
> >
> > If there are any additional points or feedback you would like us to consider, please do not hesitate to let us know. Your insights are invaluable to us, and we are eager to address any remaining issues to further improve our work.
> >
> > Thank you once again for your time and effort in reviewing our paper.

---

### Note · Authors · 2025-08-15

We sincerely thank the SACs, ACs, and reviewers for their time, constructive feedback, and thoughtful engagement throughout the review process, and we briefly summarize our work and key points addressed in the rebuttal.

**Summary of Contributions:**

We propose a novel method to directly enhance the classification performance of the weak modality, thereby achieving balanced learning between strong and weak modalities. Furthermore, we provide a theoretical analysis of the convergence properties of the cross-modal gap function, thus ensuring the effectiveness of the proposed boosting scheme.

Our contributions are listed as follows:
- Introduce a novel multimodal learning method with a sustained boosting strategy to dynamically enhance classification capability of weak modality.
- Analyze the effect of the boosting algorithm on the gap function and theoretically prove its convergence.
- Conduct extensive experiments to demonstrate the effectiveness of our method.

We believe these contributions will advance research on modality imbalance and practical multimodal learning models.

**Reviewers' Positive Feedback:**

Multiple reviewers acknowledged both the novelty and the significance of our proposed approach.
- Reviewers #Y2TP, #5gtj, and #h6gC recognized the novelty and methodological soundness of our method.
- Reviewers #Y2TP, #5gtj, #h6gC, and #3wbR acknowledged the effectiveness of our experiments, and Reviewer #5gtj further noted their completeness.
- Reviewers #Y2TP, #h6gC, and #3wbR acknowledged the theoretical support provided by our work.
- Reviewer #h6gC further emphasized the practical applicability of our method in the scenarios with three modalities.

**Addressing Reviewers' Concerns:**

During the rebuttal process, we addressed the key issues raised in the initial review, including algorithm stability in uncertain environments, model training and inference overhead, the applicability of alternative boosting strategies and early fusion architectures, score function selection, experimental details, and corrections of the typographical errors identified by the reviewers.

Reviewers #5gtj, #h6gC, and #3wbR indicated that the discussion during the rebuttal phase satisfactorily addressed their concerns and provided positive feedback. The reviewers’ comments greatly improved our paper, and all issues will be addressed in the final version. We sincerely thank the SACs, ACs, and reviewers for their time and effort.


Best regards.\
Authors of #3558.

---

### Decision · Program_Chairs · 2025-09-17

**Decision:**

Accept (oral)

**Comment:**

This paper presents a novel and well-motivated approach to addressing modality imbalance in multimodal learning by viewing it as a disparity in classification ability and proposing a sustained boosting algorithm with adaptive classifier assignment. The method is theoretically grounded, empirically validated on six datasets (including tri-modal cases), and shows consistent improvements over strong baselines, with thorough ablation and sensitivity studies. While reviewers initially raised concerns about computational overhead, modest gains in some cases, and missing evaluations, the rebuttal effectively addressed these through additional experiments and clarifications. With these issues resolved, the reviewers reached a consensus to recommend acceptance, recognizing the work’s novelty, robustness, and potential impact on robust multimodal learning.